# Natural Language Instruction-following with Task-related Language Development and Translation

**Jing-Cheng Pang**[*], **Xinyu Yang**[*], **Si-Hang Yang, Xiong-Hui Chen, Yang Yu**[†]
National Key Laboratory of Novel Software Technology, Nanjing University
Polixir Technology
{pangjc,yangxy,yangsh,chenxh}@lamda.nju.edu.cn, yuy@nju.edu.cn

## Abstract

Natural language-conditioned reinforcement learning (RL) enables agents to follow human instructions. Previous approaches generally implemented language-conditioned RL by providing the policy with human instructions in natural language (NL) and training the policy to follow instructions. In this is *outside-in* approach, the policy must comprehend the NL and manage the task simultaneously. However, the unbounded NL examples often bring much extra complexity for solving concrete RL tasks, which can distract policy learning from completing the task. To ease the learning burden of the policy, we investigate an *inside-out* scheme for natural language-conditioned RL by developing a task language (TL) that is task-related and easily understood by the policy, thus reducing the policy learning burden. Besides, we employ a translator to translate natural language into the TL, which is used in RL to achieve efficient policy training. We implement this scheme as TALAR (TAsk Language with predicAte Representation) that learns multiple predicates to model object relationships as the TL. Experiments indicate that TALAR not only better comprehends NL instructions but also leads to a better instruction-following policy that significantly improves the success rate over baselines and adapts to unseen expressions of NL instruction. Besides, the TL is also an effective sub-task abstraction compatible with hierarchical RL.

## 1 Introduction

Enabling robots to collaborate effectively with humans is a crucial aspect of machine intelligence. Natural language is a potent tool that reflects human intention and has been widely researched for instructing robot execution, designing rewards, and serving as an observation or action in reinforcement learning (RL)[1]. We are particularly interested in developing agents to follow human instructions in this broad context. Natural language-conditioned reinforcement learning (NLC-RL) is a promising approach to training instruction-following agents. It provides the policy with human instructions in natural language (NL) and trains it to follow instructions with RL algorithms. In this *outside-in* learning (OIL) scheme, the natural language instructions are directly exposed to the policy, requiring it to understand and interpret them while accomplishing RL tasks (Fig. 1-left).

However, natural language is a complex and unbounded representation of human instruction. It places an additional burden on the policy to understand diverse natural languages when solving a specific RL task. To address this issue, previous studies have attempted to convert the natural language to simpler latent embeddings [2] using pre-trained language models such as BERT [3] and GPT [4]. However, these embeddings are typically learned independently of the RL task, making capturing the task-related information in NL instructions difficult. Alternatively, some methods

---

[*]Equal contribution.
[†]Corresponding author.

37th Conference on Neural Information Processing Systems (NeurIPS 2023).

[5, 6, 7] design rule-based, task-related semantic vectors to represent NL instructions. These methods show promising results but require a laborious design of semantic vectors and have limitations in handling complex NL instructions.

This paper proposes a novel *Inside-Out* Learning (IOL) scheme for NLC-RL. As shown in Fig. 1-right, IOL automatically generates a task language (TL) that is both task-related and concise, enabling easy comprehension by the policy. IOL consists of three key components: (1) a TL generator that develops task language, (2) a translator that translates NL into TL, which facilitates policy learning in RL, and (3) a policy that solves the RL tasks. We introduce a specific implementation of IOL, called TALAR for **TA**sk **L**anguage with predic**A**te **R**epresentation. In particular, TALAR models the TL generation process as a referential game, known for its effectiveness in developing effective language. Besides, TALAR learns to model the relationships between objects as TL. Given the labelled NL to TL, the translator is trained to minimize the

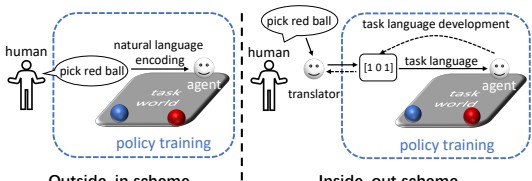

Figure 1: An illustration of OIL and IOL schemes in NLC-RL. **Left:** In OIL, the policy directly takes the (outside) NL instructions embedding as the input. **Right:** IOL first develops an (inside) TL, and policy takes TL as the input. The dashed lines represent TL development and translation, while the solid lines represent the instruction following the process.

difference between the translated TL with the target TL. Finally, arbitrary RL policy optimization algorithms can optimize the instruction-following policy.

Our paper presents several contributions to the field of NLC-RL. Firstly, we propose a novel NLC-RL scheme called IOL, which introduces a task language that enhances policy understanding and improves the learning efficiency of instruction-following. Secondly, we introduce a specific implementation of IOL that models TL development as a referential game and uses multiple learned predicates to represent TL. Thirdly, we conduct comprehensive experiments in two environments, FrankaKitchen [8], and CLEVR-Robot [9], demonstrating that the policy learned by our proposed method outperforms existing approaches in terms of its ability to follow natural language instructions and adapt to previously unseen instructions. Furthermore, the resulting TL effectively represents human instructions, providing a solid baseline for sub-task abstraction in hierarchical RL [10].

## 2   Related Work

This section begins with a summary of prior research on instruction following with RL, followed by two paragraphs discussing works pertinent to our methodology, i.e., language generation in RL and language translation.

**Instruction following with RL.** Instruction-following problems require agents to perform tasks specified by natural language instructions. A line of works solves the problems with semantics vectors [11, 12, 13], or language-conditioned policies [14, 15, 16]. To train the policies, previous methods typically process NL using a language model and train a following policy with RL algorithms. For example, [17] encodes a single-sentence instruction with a pre-trained language model and feeds the policy with the NL encoding. [18] learns a policy that maps NL instructions to action sequences by marginalizing implied goal locations. [19] combines human instructions with agent observations via a multiplication-based mechanism and then pre-trains the instruction-following policy using behaviour cloning [20]. Instead of directly converting NL to encoding, [5] encodes NL to a manually-designed binary vector in which each element has semantics meaning. Besides, the instruction-following policy has close ties to Hierarchical RL [10] because the instructions can be naturally viewed as a task abstraction for a low-level policy [21]. HAL [9] takes advantage of the compositional structure of NL and makes decisions directly at the NL level to solve long-term, complex RL tasks. These previous methods expose the unbounded NL instructions directly to the policy or encode the NL instructions to a scenario-specific manual vector, both of which have limitations. In contrast, our method automatically develops a task language, which is easily understood by the policy and facilitates policy learning.

**Language generation in RL.** Language generation plays a crucial role in developing task language. In reinforcement learning, researchers often approach this issue in a multi-agent setting, where agents

must learn effective message protocols to communicate with their partners and complete tasks. For example, the study by [22] investigates emergent language in a referential game. At the same time, a line of works has explored multi-agent communication [23, 24] for practical cooperation, considering the communication message as language. Regarding language representation, prior research has often used discrete vectors due to the discrete nature of human language. For example, [25] enables agents to communicate using discrete messages and shows that discrete representation performs similarly to continuous representation with a much smaller vocabulary size. One-hot representation [26, 22] and binary representation [5, 27] are popular forms of discrete language representation. For instance, [26] uses a one-hot language representation to enable two agents to communicate and differentiate between images. In this paper, we develop task language following the discrete form of language representation while using the predication representation.

**Language translation.** In this paper, TALAR translates natural language into task language, which lies in the domain of language translation [28]. The field of natural language processing (NLP) has extensively investigated various approaches to language translation [29, 30]. Among these approaches, the encoder-decoder architecture is a promising method because it can extract effective features from input sentences [31]. For example, the study in [32] introduces a continuous latent variable as an efficient feature for language translation, utilizing a variational auto-encoder [33]. In this paper, our primary focus is not on language translation. Therefore, we employ a more straightforward method for translation, utilizing the pre-trained BERT model as the NL encoder. In this approach, we consider NL as the source language and the target language (TL) as the target language.

## 3 Background

### 3.1 RL and NLC-RL

A typical RL task can be formulated as a Markov Decision Process (MDP) [34, 35], which is described as a tuple $\mathcal{M} = (\mathcal{S}, \mathcal{A}, P, r, \gamma, d_0)$. Here $\mathcal{S}$ represents the state space. $\mathcal{A}$ is the finite action space defined by $\mathcal{A} = \{a_0, a_1, \cdots, a_{|\mathcal{A}|-1}\}$. $P$ represents the probability of transition while $r$ represents the reward function. $\gamma$ is the discount factor determining the weights of future rewards, whereas $d_0$ is the initial state distribution. A policy $\pi : \mathcal{S} \to \Delta(\mathcal{A})$ maps state space to the probability space over action space. In NLC-RL, the agent receives an NL instruction ($L$) that reflects the human's instruction on the agent. An instruction-following policy $\pi(\cdot|s_t, L)$ is trained to make decisions based on the current state $s_t$ and NL instruction $L$. The overall objective of NLC-RL is to maximize the expected return under different NL instructions:

$$\mathbb{E}\left[\sum_{t=0}^{\infty} \gamma^t r(s_t, a_t, L) \big| s_0 \sim d_0, a_t \sim \pi(\cdot|s_t, L)\right]. \tag{1}$$

For the accuracy of sake, we use $L_\text{N}$ and $L_\text{T}$ to denote NL and TL, respectively.

### 3.2 Referential Game

Referential game (RG) has been established as a multi-agent communication task [22], which consists of two agents: a *sender* and a *receiver*. The two agents communicate using their language to identify a particular object or concept in a specific task. In a classic referential game, a pair of images is presented to the sender and the receiver, with the sender being informed which of the two images is the target. The sender must generate an effective communication message that enables the receiver to identify the target image successfully. Through interactive communication, the agents develop *meaningful* language. As the sender must capture the key feature of the task, the RG presents an ideal setting for generating task language. In TALAR, we use an extension to the basic RG to generate TL, which will be elaborated in Section 4.

### 3.3 Predicate Representation

Predicate representation utilizes discrete binary vectors to describe the relationships between objects or abstract concepts. The predicate representation is formally defined as comprising two key components: (1) *predicate* signifies the relationship between objects or concepts, which may involve one or multiple objects. (2) *argument* serves as the input for the predicate, indicating the specific objects between which the predicate measures the relationship. By employing these components,

predicate representation offers a clear and concise depiction of the relation between objects. For instance, the predicate representation vector[3] [1, 1, 0, 0, 0, 1, 0] could represent a predicate expression `Pred(a,b)`. In this case, `Pred` is a predicate that signifies a relationship, and the symbols `a` and `b` are its arguments. In the vector, the initial code [1] indicates that the value of `Pred` is True (i.e., the relationship holds). In contrast, the following red and blue one-hot codes represent the indexes of arguments `a` and `b`, respectively.

Prior research has shown that predicate representation can be learned by employing networks to output predicate values. In this manner, the learning system can automatically identify relationships in a block stacking task [36]. In TALAR, neural networks learn both predicates and their arguments. For additional discussions on predicate representation, refer to Appendix A.

### 3.4 Natural Language as Input for the Neural Network

NL sentences are variable lengths and cannot be fed directly into a fully connected network. A standard solution is to encode each word as an embedding [37] and loop over each embedding using a recurrent model. Besides the recurrent model, the pre-trained language model, e.g., Bert [3], tokenizes the words in the sentence and extracts sentence embedding based on these tokens. This process involves passing the input sentence through a series of transformer-based encoder layers that capture contextual information and create representations of the input tokens. BERT then applies a pooling layer to obtain a fixed-length vector representation of the input sentence. In this study, we employ Bert to process NL sentences to fixed-length vectors because it is lightweight and efficient.

## 4 Inside-out Learning and Implementation

This section introduces our primary contribution to efficient policy learning in NLC-RL, the IOL framework and its specific implementation, TALAR. We begin by formulating the IOL framework.

### 4.1 Inside-out Learning Formulation

We consider NLC-RL tasks, where the agent is provided with a natural language instruction as a goal to be completed. The key idea of IOL is to develop a task-related language to facilitate policy understanding. To implement IOL, three key components must be considered in IOL. We will elaborate on these components and discuss their impact on solving NLC-RL tasks.

**TL generator.** The TL generator is the critical component in IOL, designed to develop a task-related language. We expect the resulting TL to effectively and succinctly convey the task's objective to the policy. In this way, IOL mitigates the issues encountered in OIL, where the policy struggles to comprehend unseen natural language instructions. We believe that *expressiveness* and *conciseness* are vital properties of TL. Expressiveness ensures the TL accurately conveys the task goal, while conciseness promotes policy comprehension. To fulfil these requirements, we propose representing TL using predicate representation, considered both expressive and concise [38].

**Translator and Policy learning.** The translator serves as a tool that connects policy with TL, which translates NL to TL. Utilizing the translator, the policy can be optimized through various RL algorithms to improve its ability to follow human instructions. The policy learning is a goal-conditioned RL [39] problem. For complex tasks, plenty of well-established techniques in machine translation and goal-conditioned RL domains can be employed to improve policy learning in IOL further. Note that the primary focus of this paper is to investigate the effectiveness of IOL. Therefore, we employ straightforward translator and policy learning implementations, as detailed in Section 4.2.

### 4.2 Our Method

We first introduce the task dataset, which is used for TL development and translation in TALAR.

*Definition* 1 (**Task dataset**). *A task dataset $\mathcal{D} = \{(s, s', L_N)_i\}$ consists of multiple triplets. Each triplet contains a natural language instruction $L_N$ and a task state pair $(s, s')$, where $L_N$ describes state change from $s$ to $s'$ in natural language, e.g., move the red ball to the blue ball.*

---

[3]The vector could be extended to represent multiple predicate relationships simultaneously.

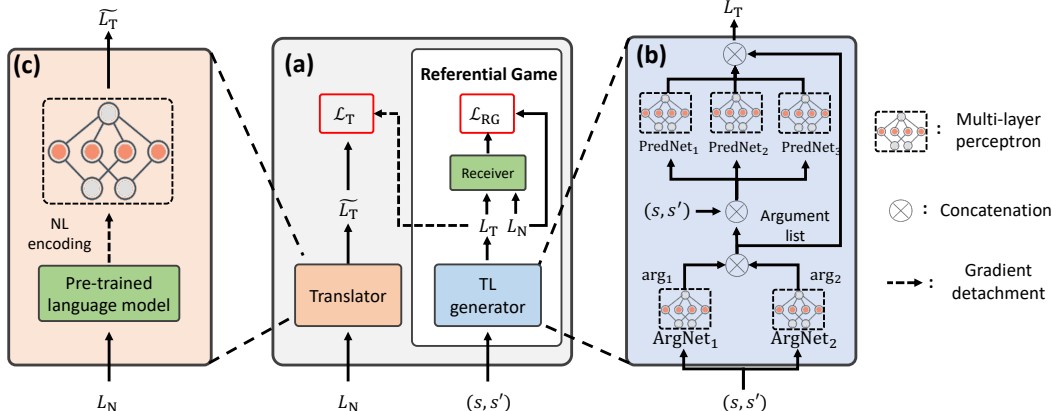

Figure 2: Overall framework of TALAR. **(a)** Overall training process: The TL generator develops TL via playing a referential game with a receiver, and the translator translates NL to TL. **(b)** Network architecture of the TL generator. **(c)** Network architecture of the translator. The number of predicate arguments and networks can be adjusted according to the task scale.

We use a state pair instead of a single state for the following reasons: (1) NL instruction typically describes a state change, e.g., turning the wheel to the left; (2) it is not straightforward to describe a single state concisely in complex task scenarios. Fig. 2 illustrates how TALAR makes use of task dataset for task language development and translation. The subsequent subsections will elaborate on training three critical components of TALAR, i.e., task language development, natural language translation and policy optimization.

### 4.2.1 Task Language Development in Predicate Representation

The TL generator denoted as $g_\theta(s, s')$, aims to develop a language in predicate representation (Section 3.3). In TALAR, the TL generator is represented by neural networks and parameterized with $\theta$. It takes a state pair $(s, s')$ as the input and outputs task language $L_T$. The We will discuss how TALAR constructs a referential game to facilitate the development of task language.

**Referential game.** The RG process is presented in the right part of Fig. 2. Based on the RG framework in Section 3.2, TALAR adapts it such that the TL generator serves as the *sender* [4] and generates the TL $L_T = g_\theta(s, s')$ to describe the state change of the given state pair. Conditioned on the generated TL $L_T$, a *receiver* aims to predict the word in the NL instruction corresponding to $(s, s')$. In this way, the TL generator must capture the critical information in the given state pair and generate effective TL, ensuring that the receiver can predict correctly. We use a pre-trained BERT model followed by a fully-connected layer to implement the receiver to predict the next token. We denote this as $f_\delta(T_i | L_T, b(T_{1,2...i-1}))$, where $f$ is the fully-connected layer after the BERT output, $b$ is the BERT model, $T_i$ is the $i$-th token of the sentence, and $\delta$ represents the parameters of the fully-connected layer. The token to be predicted $T_i$ is randomly sampled from the whole sentence. The objective for training the TL generator and receiver is to minimize the negative log-likelihood of the predicted token given the generated TL and the context tokens, which is expressed as:

$$\mathcal{L}_{\text{RG}}(\theta, \delta) = \mathbb{E}_{(s,s',L_N) \sim \mathcal{D}, T_i \sim T_{L_N}} \left[ -\log f_\delta(T_i | g_\theta(s, s'), b(T_{1,...,i-1})) \right], \quad (2)$$

where $T_{L_N}$ is the token set of $L_N$. In practice, we fix the parameters of BERT and only optimize the fully-connected layer.

**Network architecture of TL generator.** We design a specific network architecture for learning the predicate representation (we refer readers to Section 3.3 for elaborations on predicate representation). As illustrated in Fig. 2(b), the TL generator first extracts $N_a$ arguments $(\text{arg}_1, \text{arg}_2, \cdots, \text{arg}_{N_a})$ based on the input state pair, and subsequently determines the Boolean values of $N_{\text{pn}}$ predicates, given the extracted argument list. The predicate values are concatenated with the argument list and form the task language $L_T^i = (\text{pred}_1, \cdots, \text{pred}_{N_p n}, \text{arg}_1, \cdots, \text{arg}_{N_a})$. The number of predicate networks $N_{\text{pn}}$ and arguments networks $N_a$ can be adjusted according to the RL task scale.

---

[4]To avoid confusion, we will use "TL generator" instead of "sender" for the remainder of this paper.

Specifically, the arguments for the predicate are extracted by argument networks, denoted by $\text{ArgNet}_i(s, s')$. An argument network is implemented as a fully-connected network ending with a Gumbel-Softmax activation layer [40]. Through the Gumbel-Softmax, the argument network can output a discrete one-hot vector $\text{arg}_i$ in form like $(0, 1, \cdots, 0)$, which represents an abstract object or concept in the task. TALAR utilizes multiple predicate networks, denoted by $\text{PredNet}_i(s, s', \text{arg}_1, \cdots, \text{arg}_{N_a})$, to determine the Boolean values of a set of predicates. These predicates are anonymous as we do not need to pre-define them in advance. This learning manner requires predicate networks to automatically discover meaningful relationships in the task. Each predicate network outputs a 0-1 value, ending with a Gumbel-Softmax layer. All these 0-1 values are concatenated with the argument list, yielding the task language $L_\text{T}$, a discrete binary vector. Note that without the argument list in $L_\text{T}$, the resulting language cannot express different objects and therefore loses its expressiveness. The Gumbel-Softmax activation technique permits the differentiation of the entire TL generation procedure.

### 4.2.2 Natural Language Translation

The objective of the translator is to translate the natural language to the task language. TALAR uses a pre-trained BERT model to convert $L_\text{N}$ into a fixed-length embedding, followed by fully-connected layers to predict the task language corresponding to $L_\text{N}$. Fig. 2(d) presents the structure of the translator. We let $\widetilde{L_\text{T}}$ denote the TL generated by the translator, and $p_\phi$ the fully-connected layers parameterized with $\phi$. The translator is trained to minimize the difference between the predicted TL with the target TL:

$$\mathcal{L}_\text{T}(\phi) = \mathop{\mathbb{E}}_{(s, s', L_\text{N}) \sim \mathcal{D}} \left[ -\log p_\phi(g_\theta(s, s')|b(T(L_\text{N}))) \right]. \tag{3}$$

Here $T(L_\text{N})$ denotes the tokenized natural language sentence. The optimized translator trains an instruction-following policy to complete the human instructions, as described below.

### 4.2.3 Policy Optimization

TALAR uses reinforcement learning to train an Instruction-Following Policy (IFP) $\pi(\cdot|s, \widetilde{L_\text{T}})$. When the agent collects samples from the environment, the task generates a random human instruction in NL, which is then translated into the task language $\widetilde{L_\text{T}}$ by the translator. Next, the IFP makes decisions for the entire episode based on the current observation and $\widetilde{L_\text{T}}$ until completing the instruction or reaching the maximum timestep. The IFP can be optimized with an arbitrary RL algorithm using the samples collected from the environments. We use PPO [41] for TALAR and all baselines in our implementation. Note that during IFP training, the translator's parameters are fixed to prevent the translator from overfitting the current IFP.

## 5 Experiments

We conduct a series of experiments to evaluate the effectiveness of TALAR and answer the following questions: (1) How does TALAR perform compared to existing NLC-RL approaches when learning an instruction-following policy? (Section 5.1) (2) Can TALAR learn effective task language? (Section 5.2) (3) Can TL acquire any compositional structure and serve as an abstraction for hierarchical RL? (Section 5.3) (4) What is the impact of each component on the overall performance of TALAR? (Section 5.4)

**Evaluation environments**. We conduct experiments in FrankaKitchen [8] and CLEVR-Robot [9] environments, as shown in Fig. 3. FrankaKitchen aims to control a 9-DoF robot to manipulate various objects in a kitchen, involving six sub-tasks: open the microwave door, move the kettle to the top left burner, turn on the light switch, open the slide cabinet, activates the top burner and activate the bottom burner. We treat

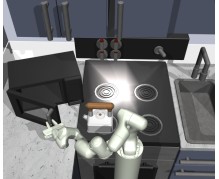 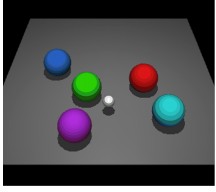

(a) FrankaKitchen     (b) CLEVR-Robot

Figure 3: A visualization of environments in our experiments. (a) FrankaKitchen. The agent controls a 9-DoF robot to manipulate various objects in a kitchen. (b) CLEVR-Robot. The agent (silverpoint) manipulates five movable balls to reach a specific goal configuration. An example of NL instruction is: *Can you move the red ball **to the left of** the blue ball?*

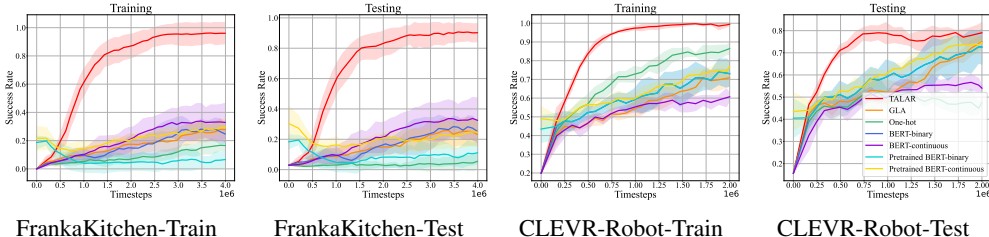

| FrankaKitchen-Train | FrankaKitchen-Test | CLEVR-Robot-Train | CLEVR-Robot-Test |

Figure 4: Training curves of different methods on two evaluation environments. The x-axis represents the timesteps agent interacts with the environment, and the y-axis represents the success rate of completing the instructions. The shaded area stands for the standard deviation over five random seeds.

each sub-task as a different goal configuration. In each trajectory, the environment randomly generates an NL instruction that describes a goal configuration. We let ChatGPT [42] generate 50 different NL instructions to describe each goal configuration, yielding 300 different NL instructions in this task. CLEVR-Robot is an environment for object interaction based on the MuJoCo physics engine [43]. It contains five movable balls and an agent (silver point). In each trajectory, the agent aims to achieve a goal configuration that requires moving a specific ball in a target direction (front, behind, left, or right) relative to a target ball. An example of NL instruction is *Move the green ball **to the left of** the blue ball.* There are 80 different goal configurations in total. We use 18 natural language sentence patterns to describe each goal, resulting in 1440 distinct NL instructions. We split the NL instructions into two tasks: training and the testing set. The training set comprises 40 and 9 different NL instructions for each goal configuration in FrankaKitchen and CLEVR-Robot, respectively, resulting in a total of 240 NL instructions for FrankaKitchen and 720 for CLEVR-Robot. During training, the agent can only interact with the NL instructions in the training set. The testing set consists of the remaining NL instructions, which include 120 NL instructions for FrankaKitchen and 720 for CLEVR-Robot. Refer to Appendix D.1 for more details about the environments and the NL instructions in the training and testing sets.

**Task dataset collection**. The task dataset can be obtained from pre-collected data, where state pairs are annotated with natural language instructions through human input or rule-based functions. In our experiment, we utilize PPO to train a goal-conditioned policy capable of achieving any goal configuration. We then use the trained policy to collect trajectory data $(\{s_0, a_0, s_1, a_1 \cdots s_T\}, L_N)$ across various environments, where $L_N$ is the natural language that describes the goal configuration of this trajectory and is randomly sampled from the training instructions set. The collected trajectory data is rearranged, with $\{(s_i, s_T, L_N)\}_{i=0}^{T-1}$ as tuples in the task dataset, resulting in 50,000 tuples for the FrankaKitchen task, and 100,000 for the CLEVR-Robot task. Note that the collected trajectory data can be used to pre-train the policy. With tuple data such as $(s_i, a_i, L_N)$, the policy $\pi(s, L_N)$ can be pre-trained to output the action $a$. Two baselines (Pre-trained BERT-* in Section 5.1) utilize this data for pre-training purposes.

**Implementation Details.** For experiments, we utilize the pre-trained BERT-base-uncased model[3] as the NL encoder. We conduct all experiments with five random seeds, and the shaded area in the figures represents the standard deviation across all five trials. We refer readers to Appendix D for more implementation details about the experiments.

## 5.1 Performance of Instruction-following Policy

**Baselines for comparison.** We consider multiple representative methods in NLC-RL as baselines: (1) **One-hot** encodes the representation of all possible natural language instructions (including training and testing) to a one-hot vector, where each instruction is assigned a unique position. (2) **BERT-binary** processes the natural language with a pre-trained BERT model, and the resulting language encoding is then transformed into a binary vector using a fully-connected network. This binary vector's size equals the TL generated by TALAR. To ensure differentiability, we employ a reparameterization trick [44] that converts a continuous vector into a binary vector. (3) **BERT-continuous** is similar to BERT-binary, except it uses a continuous vector of the same size. (4) **Pre-trained BERT-binary/continuous** initially pre-trains the policy networks and the BERT model using behavior cloning [45] with pre-collected $(s_i, a_i, L_N)$ tuples, and subsequently trains the policy networks with RL. The data collection process is described above alongside the task dataset collection. (5) **GLA** (Grounding Language to latent Action) [46] first employs an auto-encoder to learn a latent

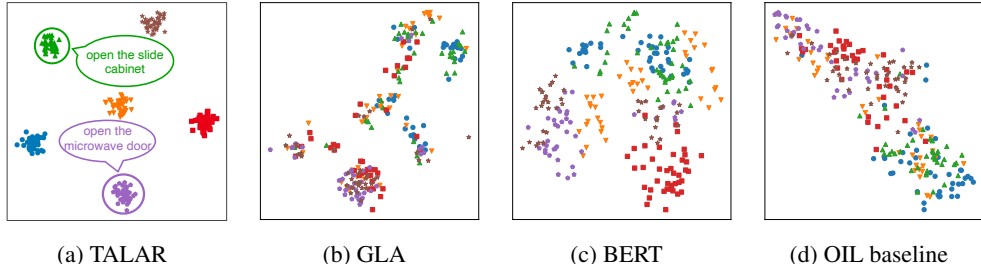

| (a) TALAR | (b) GLA | (c) BERT | (d) OIL baseline |

Figure 5: The t-SNE projections of different NL encoding types on FrankaKitchen. Points with the same marker encode forty different NL expressions that describe the same goal configuration. We add a slight noise to the overlapping points for better presentation. **(a)** The translator's t-SNE representations of the TL output. **(b)** The encoding output by the language encoder in GLA. **(c)** The encoding output by the NL encoding layer of the BERT model. **(d)** The encoding output by the NL encoding layer of the Pre-trained BERT-continuous.

action embedding and then encodes the NL into the learned latent action space, which is a binary vector with the same size as TL.

**Experimental results.** Fig. 4 presents the training curves of the instruction-following policies on different natural language instruction datasets. In general, TALAR achieves a superior instruction-following policy that enhances the success rate by 63.2% on FrankaKitchen and 13.4% on CLEVR-Robot compared to baseline method and generalizes well to the NL instruction expressions that were not previously encountered. On the FrankaKitchen task requiring more difficult robot manipulation, TALAR significantly outperforms all baselines. This result demonstrates the advantage of IOL, which eases the policy's burden on understanding natural language. On the training NL instruction set, TALAR achieves a high success rate within a reasonable number of timesteps, significantly faster than the other baselines. Additionally, TALAR demonstrates a remarkable success rate on testing sets in both environments, showcasing greater capacity than the baselines. Although One-hot performs adequately on the training NL set, its ability to generalize to the testing NL sets is limited. BERT-based baselines, built on the OIL framework, improve slower than TALAR on the training NL instruction set. In addition, the NL instructions on the FrankaKitchen task are typically more complex. OIL methods become increasingly more challenging to achieve a high task success rate.

## 5.2 Analysis on the Generated Task Language

We further investigate the reasons behind TALAR's more efficient learning compared to baseline methods. To explore this issue, we employ the optimized translator to convert NL instructions into TL and project the resulting TL onto a two-dimensional plane using t-SNE [47]. The projection results are displayed in Fig. 5a. The results show that the learned TL effectively distinguishes NL instructions representing different goal configurations, developing a unique representation for each configuration. This conclusion is supported by the fact that various NL expressions for the same goal configuration translate into similar, or even identical, TL. Consequently, TALAR can easily recognize these expressions and concentrate on learning how to complete the RL task. For comparison, we also project the NL encoding output generated by other baseline methods, as illustrated in Fig. 5(b-d). The NL encoding output from these baseline methods exhibits greater diversity compared to TALAR. These results suggest that the OIL baseline treats diverse NL expressions for the same task objective as different goal configurations, potentially distracting from RL and slowing policy learning. It is worth noting that BERT, an established pre-trained model for natural language understanding, also fails to generate unique NL encoding. Even when fine-tuned on specific tasks, the output encoding of the BERT model is divergent, as shown in Fig. 5d.

## 5.3 TL Is an Effective Abstraction for Hierarchical RL

Previous experiments have demonstrated that the resulting TL can compactly and uniquely represent the NL instructions, facilitating policy learning in the following instruction. In this section, we further explore whether the TL learns any compositional structure by examining if it can serve as an effective goal abstraction for hierarchical RL. Specifically, we train a high-level policy outputting a TL, instructing the IFP to complete a low-level task. We define a long-term task as one in which the

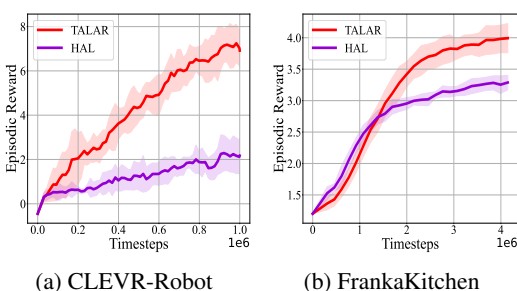
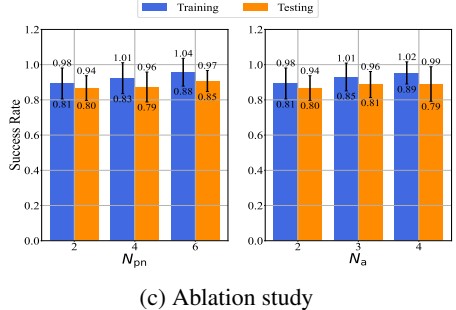

|  (a) CLEVR-Robot | (b) FrankaKitchen | (c) Ablation study |

Figure 6: Experiments of long-horizon tasks on different environments. The agent obtains a reward of +1 when completing one goal configuration.

Figure 7: Ablation study on the number of predicate and argument networks.

agent must achieve multiple goal configurations simultaneously. For CLEVR-Robot, we constructed the long-term task following the approach outlined in [9]. In this task, the agent's objective is to arrange objects to satisfy all ten goal configurations that are implicitly contained within it. For FrankaKitchen, the agent is required to complete all six sub-tasks at the same time. Refer to Appendix D.1.3 for additional details regarding the long-term tasks and the hierarchical RL setting.

For comparison, we consider a baseline method called **HAL** [9], which leverages the compositional structure of NL and directly acts at the NL level. Following HAL's original implementation, the high-level policy outputs the index of the NL instruction rather than the NL words. To ensure a fair comparison, we align all other modules except for the high-level policy of both methods, e.g., using the same IFP trained by TALAR as the low-level policy. Fig. 6 presents the comparison results. The high-level policy that uses TL as a low-level goal abstraction outperforms that using NL in terms of sample efficiency. This result demonstrates that TL can be an effective goal abstraction naturally compatible with hierarchical RL. In practice, the TL can be generated with a task dataset and serves as a sub-task abstraction for high-level policy.

## 5.4 Ablation Study

We conduct experiments to investigate how the performance of TALAR is affected by varying the number of predicate networks and the number of arguments (i.e., $N_{pn}$ and $N_a$). Specifically, we choose $N_{pn}$ from the set $\{2, 4, 6\}$ and $N_a$ from $\{2, 3, 4\}$. We trained the IFPs for 4 million timesteps on the FrankaKitchen environment. The experiment results are presented in Fig. 7. In general, increasing $N_{pn}$ and $N_a$ leads to improved performance of the IFP on both the training and testing sets. This result can be attributed to the enhanced representation abilities of the resulting TL, which enables more efficient learning by the policy. However, it is worth noting that excessively high values of $N_{pn}$ and $N_a$ may lead to reduced training efficiency due to the simultaneous learning of multiple networks. $N_{pn} = 6$ and $N_a = 4$ are sufficient for the robot manipulation tasks like FrankaKitchen.

## 6 Conclusion and Future Work

In this study, we introduce a novel learning scheme for NLC-RL, called IOL, which develops task-related language to facilitate policy learning. Our experiments demonstrate that IOL is a promising approach for efficiently training instruction-following agents. However, there are limitations in our proposed implementation of IOL, TALAR. Firstly, TALAR develops the task language using a static task dataset and, therefore, can not be directly applied to the open-world environment, where the NL instructions could vary hugely from the dataset. It is possible to mitigate this issue by dynamically extending the task dataset and fine-tuning the TL generator/translator during the policy learning process. Additionally, TALAR requires a manual reward function for policy training, which may be inaccessible if the reward design is complex. Fortunately, there have been well-validated methods for solving sparse reward problems [48, 49, 50], an effective substitute for the manual reward function. Furthermore, incorporating basic properties of predicate relationships (such as transitivity, reflexivity, and symmetry) when training the TL generator could make the resulting TL more meaningful and self-contained. Lastly, TALAR implements the translator as a fully-connected layer with a pre-trained BERT model, which may be inadequate for more complex natural language instructions. This

limitation could be addressed by employing other Natural Language Processing (NLP) literature techniques, such as semantic parsing.

Recently, there has been a growing interest in building instruction-following agent with LLMs as the *brain* [51, 52, 53]. Typically, LLMs function as planners, while a low-level policy is responsible for executing the planned tasks. However, the separation between the LLM planner and the task execution process may impede the overall task completion rate. The challenge of managing increasingly complex language instructions in open environments remains an unresolved issue. Furthermore, it is worth investigating whether LLMs can serve as effective task language generator. We hope that future research will explore these intriguing questions and contribute to developing agents that interact with humans more effectively.

## Acknowledgements

This work is partly supported by the National Key Research and Development Program of China (2020AAA0107200) and the National Science Foundation of China (61921006). The authors extend their appreciation to Wang-Zhou Dai and Yuxuan Huang for their valuable discussions on predicate representation, Xinchun Li and Yikai Zhang for their insights on language model training, Jingxuan Han for experiment details, Lei Yuan for discussing the potential applications of TALAR, and the anonymous reviewers for their useful feedback on this study.

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

# Appendix

## Table of Contents

## A  Discussion about Predicate Representation

A crucial aspect of IOL is how the task language is represented. We believe that *expressiveness* and *conciseness* are essential properties of language representation. Expressiveness ensures that the task language accurately reflects the goal configuration, while conciseness promotes ease of policy comprehension. To fulfil these requirements, we implement IOL by representing task language with predicate representation, which is considered both expressive [38] and concise as a discrete representation. The expressiveness of predicate representation stems from two key advantages: *compositional structure* and *interpretability*.

We first discuss the **compositionality**, which refers to the ability of a language to denote novel composite meanings. For example, if a language can represent *blue circle* and *red square*, it can represent *blue square* as well. This language has a compositional structure. The compositionality is seen as a fundamental feature of natural language and a pre-condition for a language to generalize at scale [55]. Predicate representation naturally has compositional structure since it can denote the composite meanings by changing the truth value of predicates.

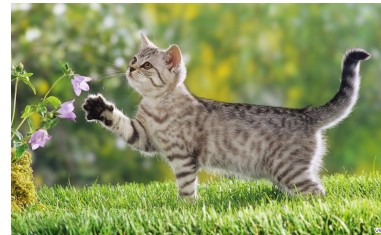

Next, we talk about the **interpretability**. Predicate representation uses multiple predicate expressions, such as `Pred(a1,a2)`, to describe the relationships in a specific environment. Previous works have found that, even if the predicate symbols are machine-generated by a learning system and anonymous to humans, they still offer *some extent of interpretability*. For example, if a learning system generates a predicate expression `Pred(cat, grassland)=True` for Fig. 8, we can guess that `Pred` means [stand] or [above]. When more figures and predicate expressions are provided, the actual meaning of these anonymous predicate symbols would be more apparent to humans. The primary focus of this paper is to explore efficient learning methods, and as such, we do not delve into the topic of interpretability.

Figure 8: An illustration of the interpretability of predicate representation. For an anonymous predicate expression: `Pred(cat, grassland)=True`, we can guess that `Pred` represents [above].

In conclusion, we propose to use predicate representation to represent task language since nearly all tasks contain relationships. For instance, an Atari game [56] includes a relationship between the agent and the antagonist, a first-person shooter game contains a relationship between the gun and the bullets, a stock trading task contains a relationship between stocks and price, etc. Even if some tasks are highly abstract and have no apparent relationships, we can implement the IOL by replacing predicate representation binary or continuous vectors.

## B  Additional Related Work

**Hierarchical Reinforcement Learning** (HRL) approaches are promising tools for solving complex decision-making tasks [57]. HRL solves the problem by decomposing it into simpler sub-tasks using a hierarchy of policies learned by RL [58, 59, 60]. For example, to put a drink into a fridge, you will (1) take up the drink, (2) open the fridge door, (3) put down the drink, and (4) close the fridge door. Putting a drink into a fridge is a long-term task, while these four are sub-tasks. Typically, a high-level policy is trained to decompose the main task into sub-tasks (three steps in the example), and low-level policies (or single policy) are trained to complete these sub-tasks. A *core problem* in HRL is how to represent the goal of the sub-task (i.e., goal representation) that the low-level policy follows. Previous works have investigated both concrete and abstract goal representations of a sub-task. Concrete goal representation can be the *target position* to reach [61], or a *target state* [62]. As for abstract goal representation, it can directly be natural language [9] or an encoding of action primitives [64]. Back to the topic of this paper, language is a natural goal representation for the sub-tasks, such as *open the fridge door* in the above example. Through our experiments (Section 5.3), we observe that the resulting TL is an effective abstraction of the goal of the sub-task, which demonstrates the applicability of TL to work together with HRL.

# C  Algorithm Descriptions

Algorithm 1, 2, and 3 present the training procedures of the TL generator, the translator and the instruction-following policy, respectively.

---

**Algorithm 1** Training procedure of the TL generator.

---

**Input**: task dataset $\mathcal{D} = \{(s, s', L_{\mathrm{N}})_i\}$, pre-trained BERT model $b$.
**Output**: the optimized TL generator.

1: Initialize the TL generator $g_\theta$.
2: **while** training not complete **do**
3:     Sample a batch of data from $\mathcal{D}$.
4:     Update $\theta$ to minimize the RG loss (Eq.(2)).
5: **end while**
6: **return** the optimized TL generator $g_\theta$.

---

**Algorithm 2** Training procedure of the translator.

---

**Input**: task dataset $\mathcal{D} = \{(s, s', L_{\mathrm{N}})_i\}$, the optimized TL generator $g_\theta$, the translator network $p_\phi$ parameterized with $\phi$, and the pre-trained BERT model.
**Output**: the optimized TL translator.

1: Initialize the translator $t_{\phi_1, \phi_2}$ with parameters $\phi_1$ and $\phi_2$.
2: **while** training not complete **do**
3:     Sample a batch of data $\{(s, s', L_{\mathrm{N}})_j\}$ from $\mathcal{D}$.
4:     // Compute the target task language.
5:     Calculate the task language $L_{\mathrm{T}} = g_\theta(s, s')$.
6:     // Optimize the translator.
7:     Update $\phi$ to minimize the loss in Eq.(3).
8: **end while**
9: **return** the optimized translator $p_\phi$.

---

**Algorithm 3** Training procedure of the instruction-following policy.

---

**Input**: the optimized translator $p_\phi$.
**Output**: the optimized instruction-following policy.

1: Initialize the policy function $\pi$, and the value function.
2: **while** training not complete **do**
3:     Sample a NL instruction $L_{\mathrm{N}}$ from the environment.
4:     Generate corresponding task language $\widetilde{L_{\mathrm{T}}} = p_\phi(L_{\mathrm{N}})$.
5:     // Collecting samples
6:     **while** episode not terminal **do**
7:         Observe current state $s_t$.
8:         Execute action $a_t \sim \pi(\cdot | s_t, \widetilde{L_{\mathrm{T}}})$, and receive a reward $r_t$ from the environment.
9:     **end while**
10:    // Training
11:    Update the policy and value functions with the RL algorithm.
12: **end while**
13: **return** the optimized policy $\pi$.

---

# D  Implementation Details

In our experiments, we utilize the open-sourced RL repository, stable-baselines3 [65], to implement the RL optimization. The parameters of the BERT model are fixed if not otherwise specified. All experiments are run five times with different random seeds. We will now introduce two experimental environments, followed by a detailed explanation of the model architecture of our method and the hyper-parameters utilized in our experiments.

### D.1 Tasks for Evaluation

### D.1.1 FrankaKitchen

**Task descriptions** As depicted in Fig. 3, the FrankaKitchen environment emulates a realistic kitchen setting with objects that can be interacted with, including a 9-DoF Franka robot, a microwave, a kettle, an overhead light, cabinets, and an oven. The observation space is defined as $\mathbb{R}^{30}$, representing the state of each object. The action space is $\mathbb{R}^9$, with 7 DoF for the arm and 2 for the gripper. The actions are represented as the joint velocity. At each timestep, the reward is calculated as the difference between the distance to the target robot's posture at the previous step and the distance to the target robot's posture at the current step. A reward of +1 is given for success, and a 0 is provided for failure.

**Natural language instructions.** In FrankaKitchen, six distinct goal configurations are related to operating various objects. To generate natural language instructions that describe these goal configurations, we employ ChatGPT to produce everyday language sentences. For example, we prompt ChatGPT with: "I want to ask my robot to open the microwave door. Can you provide me with 50 different instructions I can use?" We obtain 6 x 50 = 300 unique natural language instructions in this manner. To conserve space, we only present 6 x 20 (i.e., 10 in the training dataset + 10 in the testing dataset) = 120 natural language instructions used in our experiments.

- —(**Training set**)—
- Please open the door of the microwave.
- Can you open the microwave door for me?
- I would appreciate it if you could open the microwave door.
- Can you open the microwave door at your earliest convenience?
- Would you be able to open the door of the microwave?
- I need the door of the microwave to be opened, please.
- Please, could you open the door of the microwave?
- I require you to open the door of the microwave.
- Could you open the microwave door when you get a chance?
- Would it be possible for you to open the door of the microwave?
- Could you fill the kettle with water and place it on the stove?
- It's tea time! Could you turn on the stove and place the kettle on top?
- I need some hot water. Could you fill the kettle, put it on the stove, and turn on the burner?
- Please prepare some hot water by placing the kettle on the burner and turning on the stove.
- Can you fill the kettle, put it on the stove, and turn on the burner to heat up the water?
- Hey robot, can you prepare some hot water by putting the kettle on the burner and turning on the heat?
- Can you please put the kettle on the stove, robot?
- Robot, it's tea time! Please put the kettle on the burner and heat up the water.
- Hey robot, could you heat up the water in the kettle on the burner?
- I'm feeling chilly, robot. Can you put the kettle on the stove and boil some water?
- Could you illuminate the room for me?
- I need some light, can you help me?
- Can you activate the lighting system?
- Light up the room, please.
- I want to see better, please turn on the lights.
- Can you switch on the room's lighting, please?
- Can you brighten up the space?
- I need some light in here, can you turn on the lights?
- I need some illumination, can you help me?
- Would you mind me switching the light?
- Please open the cabinet door on the sliding mechanism right away.
- Please open the cabinet door by sliding it.

- I would appreciate it if you could open the sliding cabinet door.
- Please slide the cabinet door open as quickly as possible.
- Please open the sliding cabinet door promptly.
- Would you mind opening the slide cabinet door?
- Please open the cabinet door by sliding it as soon as possible.
- Can you slide open the cabinet door quickly, please?
- Can you please open the cabinet door by sliding it now?
- I need your help, open the slide cabinet door please.
- Could you kindly activate the top burner by turning the oven knob?
- Twist the oven knob to activate the top burner.
- Activate the top burner by turning the oven knob clockwise.
- Kindly turn the oven knob to activate the top burner.
- Please turn the oven knob to switch on the top burner.
- The top burner is activated.
- The top burner can be activated by turning the oven knob to the right.
- Activate the top burner by turning the oven knob to the right on the stove.
- Please turn the oven knob to switch on the top burner on the stove.
- Turn the knob on the oven that corresponds to the top burner.
- Please turn the knob on the oven that activates the bottom right burner.
- Could you turn the knob on the oven so that the bottom right burner activates?
- Would you kindly turn the knob on the oven so that the bottom right burner activates?
- Activate the bottom right burner by turning the knob on the oven, Robot.
- Please turn the oven knob to activate the bottom right burner, Robot.
- Would you mind activating the bottom right burner on the oven by turning the knob?
- Can you turn the oven knob to activate the bottom right burner, Robot?
- Robot, please activate the bottom right burner on the oven by turning the knob.
- Manipulate the oven knob to turn on the burner at the bottom right.
- Turn the oven knob to activate the bottom burner.
- —(**Testing set**)—
- Can you please open the door of the microwave?
- I need the microwave door to be opened.
- Could you open the door of the microwave quickly?
- Can you open the microwave door quickly?
- I'd like you to open the microwave door for me, please.
- Would you mind opening the door of the microwave for me?
- Can you assist me by opening the microwave door?
- Can you kindly open the door of the microwave?
- Can you please open the microwave door without delay?
- Could you please open the door of the microwave?
- Can you start heating up the water by placing the kettle on the burner and turning on the stove?
- Please start boiling some water in the kettle on the burner, robot.
- Robot, could you put the kettle on the burner and heat up the water?
- Let's make some tea. Can you put the kettle on the burner and turn on the heat?
- Can you please heat up some water by putting the kettle on the burner and turning on the stove?
- Please heat up the kettle by putting it on the burner.
- Please put the kettle on the burner, robot. I'm craving a cup of tea.
- I need some hot water. Can you put the kettle on the stove and turn on the burner?
- Can you please boil some water by putting the kettle on the burner and turning on the stove?

- I'm craving some tea. Can you put the kettle on the burner and turn on the heat?
- Please, illuminate the space.
- Turn on the lights, please.
- Could you turn on the lighting system?
- I need some illumination, can you turn on the lights?
- Can you switch on the lights in here, please?
- Please, switch on the lights.
- Please, turn on the lights in this room.
- Can you brighten up the room?
- Please switch on the light.
- Can you brighten up this space for me?
- Please slide the cabinet door open.
- Please slide the cabinet door open now.
- Can you open the cabinet door by sliding it without delay?
- Could you please open the cabinet door by sliding it?
- Can you open the sliding cabinet door right away?
- Could you slide the cabinet door open?
- Please, would you mind me opening the slide cabinet door.
- I need you to open the sliding cabinet door.
- Can you open the sliding door on the cabinet, please?
- Could you please slide the cabinet door open?
- Twist the oven knob to turn on the top burner.
- Manipulate the oven knob to activate the top burner.
- Turn the oven knob to the right to activate the top burner on the stove.
- Could you please turn the oven knob to activate the top burner?
- Use the oven knob to activate the top burner on the stove.
- Please turn the oven knob to activate the top burner.
- Turn the oven knob clockwise to activate the top burner on the stove.
- Use the oven knob to switch on the top burner on the stove.
- Can you turn on the top burner by twisting the oven knob?
- Turning the oven knob will switch on the top burner.
- Would you mind turning the oven knob to activate the bottom burner?
- The bottom burner is activated.
- Rotate the knob located on the oven to start the burner at the bottom right.
- Please turn the oven knob so that the bottom right burner activates, Robot.
- Please activate the bottom right burner on the oven by turning the knob, Robot.
- Would you mind turning the oven knob to activate the bottom right burner, Robot?
- Activate the bottom right burner on the oven by turning the knob, please.
- Turn the knob on the oven in order to activate the bottom right burner.
- Adjust the oven knob to the position that turns on the burner at the bottom right.
- Turn the oven knob to activate the bottom right burner, Robot.

In each trajectory, a natural language instruction is randomly selected, after which the agent makes decisions to complete the given instruction. This experiment aims to not only confirm the effectiveness of TALARbut also demonstrate its robustness in handling diverse and realistic scenarios.

### D.1.2 CLEVR-Robot

**Task descriptions.** As depicted in Fig. 3, The CLEVR-Robot environment requires the agent to manipulate five balls in accordance with human instructions. The observation space is $\mathbb{R}^{10}$, which represents the location of each object, and $|\mathcal{A}| = 40$ corresponds to selecting and pushing an object in one of the eight cardinal directions. At each timestep, the reward is calculated as the difference

between the distance to the target position at the previous step and the distance to the target position at the current step, with a reward of +100 for success and -10 for failure. Some previous OIL-based methods attempt to train the IFP in a sparse reward environment [66, 67]. We relax the sparse reward constraint to dense reward for two reasons: (1) Sparse reward is not the point of this study. We are more concerned with exploring a new learning scheme for NLC-RL than with resolving the sparse reward problem; (2) The applicability of current methods. Most RL approaches address the sparse reward problem using HER [48], which relabels the origin goal in the trajectory with the actual achieved state. Nonetheless, the relabeling procedure is contingent on the transformation from the state to the goal, which necessitates additional human annotation or program design.

**NL instructions.** We use eighteen different NL sentence patterns to express each goal configuration. For example, if we take a goal configuration such as "red ball **in front of** the blue ball", its corresponding NL instructions (i.e., eighteen NL sentence patterns) can be one of the following:

- —(**Training set**)—
- Push the red ball **in front of** the blue ball.
- Can you push the red ball in front of the blue ball?
- Can you help me push the red ball in front of the blue ball?
- Is the red ball in front of the blue ball?
- Is there any red ball in front of the blue ball?
- The red ball moves in front of the blue ball.
- The red ball is being pushed in front of the blue ball.
- The red ball is pushed in front of the blue ball.
- The red ball was moved in front of the blue ball.
- —(**Testing set**)—
- Move the red ball in front of the blue ball.
- Keep the red ball in front of the blue ball.
- Can you move the red ball in front of the blue ball?
- Can you keep the red ball in front of the blue ball?
- Can you help me move the red ball in front of the blue ball?
- Can you help me keep the red ball in front of the blue ball?
- The red ball is being moved in front of the blue ball.
- The red ball is moved in front of the blue ball.
- The red ball was pushed in front of the blue ball.

There are two kinds of datasets: training and testing. In the training set, goal configurations are expressed using nine NL sentence patterns, while the remaining nine NL sentence patterns are used in the testing set. At the start of each trajectory, the environment randomly samples a goal configuration and an NL sentence pattern to express the human instruction. By constructing these two datasets, we simulate the scenario in an open environment where different individuals instruct the robots using their linguistic preferences.

### D.1.3   Hierarchical RL.

**Hierarchical RL setting** The observation space of the high-level policy is the same as that of the low-level policy, i.e. IFP, resulting in $\mathbb{R}^{10}$ for CLEVR-Robot and $\mathbb{R}^{30}$ for FrankaKitchen. Our method utilizes TL space as the action space, while HAL uses NL instruction space. The action output by the high-level policy is then treated as the goal configuration of the IFP. We roll out the IFP for ten steps for every high-level action. We use sparse reward to verify the effectiveness of the high-level goal abstraction, i.e., the goal configuration assigned to IFP. The reward is calculated at each timestep as the count of the goal configurations achieved at the current timestep.

**CLEVR-Robot** As demonstrated in Fig.9a, the object arrangement task was proposed by [9], which aims to rearrange the objects in the environment to satisfy all ten implicit constraints. At the beginning of a trajectory, the environment resets the position of all balls to a random location. Following [9], the precise arrangement constraints are: (1) red ball **to the right of** purple ball. (2) green ball **to the right of** red ball. (3) green ball **to the right of** cyan ball. (4) purple ball **to the left of** cyan ball. (5) cyan ball **to the right of** purple ball. (6) red ball **in front of** blue ball. (7) red ball **to the left of** green

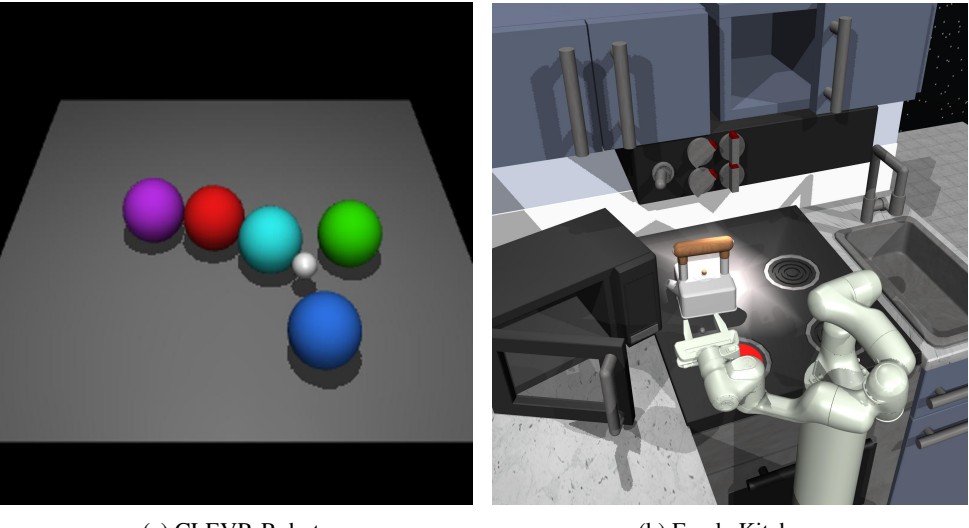

(a) CLEVR-Robot        (b) FrankaKitchen

Figure 9: Visualization of completed long-term tasks in the high-level environment.

ball. (8) green ball **in front of** blue ball. (9) purple ball **to the left of** cyan ball. (10) blue ball **behind** the red ball.

**FrankaKitchen** As demonstrated in Fig.9b, the multitask environment was introduced by [8], which aims to complete all six goal configurations. At the beginning of a trajectory, the environment resets all sub-tasks to their initial state, as shown in the leftmost column of Fig. 18.

## D.2 Network architecture

To implement TALAR, we employ the IFP network architecture illustrated in Fig. 10. Both the policy network and value network share the same architecture, but they have different output dimensions. In our experiments, all baselines utilize the same IFP network architecture as TALAR. The primary distinction lies in how these baselines encode natural language instructions. The natural language encoding for baselines is the same size as TL, and the concatenated vectors (i.e., [state, NL encoding]) serve as input for the policy and value networks.

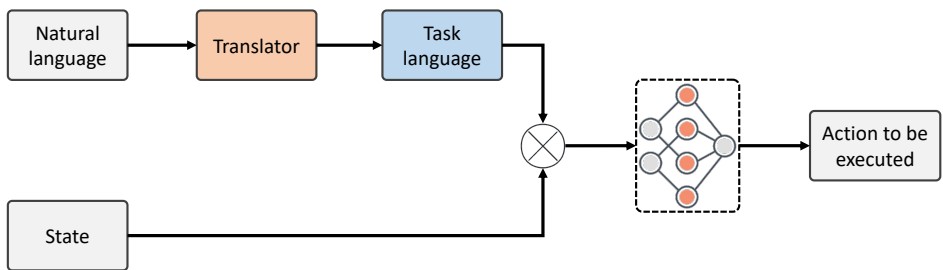

Figure 10: Network architecture of instruction-following policy network in TALAR.

## D.3 Hyper-Parameters

The hyper-parameters for implementing TALAR are presented in Table 2. When implementing baseline methods, we use the same hyper-parameters of PPO for policy learning.

Table 2: Hyper-parameters.

| Hyper-parameters | Value |
| --- | --- |
| $N_a$ | 2 |
| $N_{pn}$ | 4 |
| Size of ArgNet's output | 5 |
| Learning rate (LR) for $\mathcal{L}_{RG}$ | 3e-4 |
| LR for $\mathcal{L}_T$ | 3e-4 |
| MLP networks in translator | [256, 256, 256, 256, $|L_T|$], relu |
| Predicate network | [128, 128, 2], relu |
| Argument network | [128, 128, 5], relu |
| PPO epoch | 10 |
| PPO policy LR | 3e-4 |
| PPO value LR | 3e-4 |
| PPO policy network | [32, 64, 64], tanh |
| PPO value network | [32, 64, 64], tanh |
| PPO mini-batch size | 128 |
| PPO nums of mini-batch | 160 |
| GLA action encoder net | [2$|\mathcal{S}|$,128,128,64], relu |
| GLA action decoder net | [64,128,128,$|\mathcal{S}|$], relu |

# E    Additional Experiment Results

## E.1    Training IFP with Different Number of NL Sentence Patterns

We evaluate the robustness of TALAR by training an instruction-following policy under varying numbers of natural language sentence patterns. Specifically, we utilize the CLEVR-Robot environment and train the IFP with NL patterns number from the set $\{1, 5, 9\}$. Table 3 displays the experimental results. As the number of sentence patterns increases, TALAR consistently maintains a near 100% success rate. This is due to TALAR's ability to uniquely represent NL expressions across different sentence patterns, allowing the IFP to comprehend instructions more easily and learn to execute tasks efficiently. These results suggest that developing a task-related language is beneficial for policy learning. In contrast, the performance of baseline methods declines significantly as the number of NL sentence patterns increases, as they must simultaneously understand a greater variety of NL expressions and acquire the skills to execute tasks.

Table 3: A summary of the final success rate (%) on the training set with a different number of NL sentence patterns. Each IFP is trained for 2M timesteps on CLEVR-Robot and evaluated for 40 episodes. The results are averaged over 5 seeds.

| Method        Pattern nums. | 1 | 5 | 9 |
|---|---|---|---|
| TALAR | **100%** | **100%** | **99.9%** |
| One-hot | 98.2% | 91.8% | 86.5% |
| BERT-binary | 82.0% | 60.9% | 64.0% |
| BERT-continuous | 95.3% | 63.1% | 60.7% |

## E.2    Complete Results of T-SNE Projection of Different Representations

Figure 11 presents the t-SNE projection results of various representations on both training and testing natural language instruction datasets. Upon analyzing the training dataset of natural language instructions, it is obvious that the t-SNE projections of the task language exhibit a higher degree of clustering compared to the baselines. This observation implies that the IOL method effectively learns a concise task goal representation, which is suitable for solving NL instruction following task. The BERT-binary representation exhibits a certain capacity to distinguish natural language instructions. Nevertheless, all the data points tend to get close, suggesting a less pronounced separation of different NL instructions in comparison to the task language. In the testing dataset, the task language representation surpasses other baselines, as data points with the same marker display a higher concentration. This result suggests that the task language is more robust against unseen natural language expressions. On the other hand, the t-SNE projections of the baselines appear scattered across the plane, which imposes a natural language understanding challenge on policy learning.

## E.3    Aligning the network architecture of baselines with TALAR

In this section, we align the network structures of several baselines with TALAR in order to eliminate the impact of the additional networks on performance in TALAR. More specifically, we have added new BERT-based baselines on FrankaKitchen task, which employ the same architecture as the TL generator in TALAR, adhering to the BERT model. Other experimental settings remain same with the experiments in the paper. These new baselines are denoted with the prefix "Aligned", and their respective experimental results are presented in the Fig. 12. The performance of the baseline improves with the implementation of a new network architecture. However, it remains inferior to the TALAR approach in terms of learning efficiency and convergence scores. This highlights the effectiveness of the TALAR method.

## E.4    Ablation study on the size of dataset

We recognize that the size of task dataset is a significant factor to consider. Thus we have conducted additional experiments to investigate the impact of task sample number on the performance of the

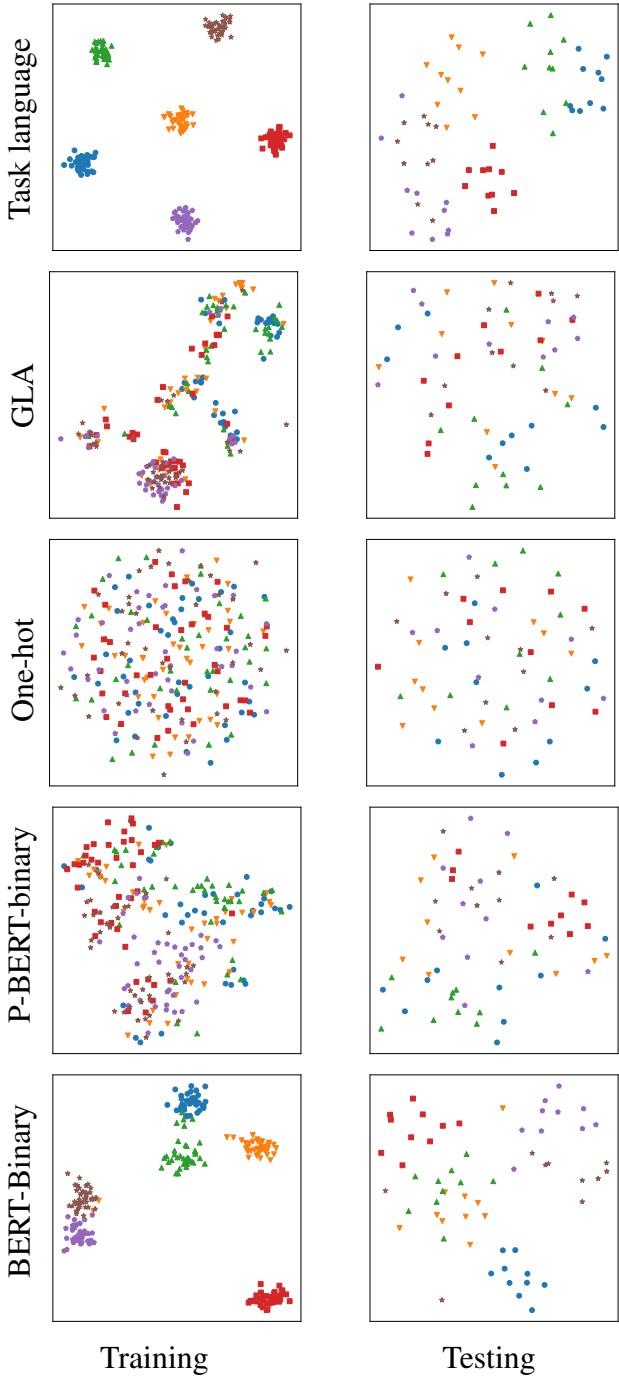

Figure 11: The t-SNE projection of different representations on two NL expressions datasets. Each row represents one kind of representation, and each column stands for different NL instruction datasets. Points with the same marker encode nine different NL expressions that describe the same goal configuration.

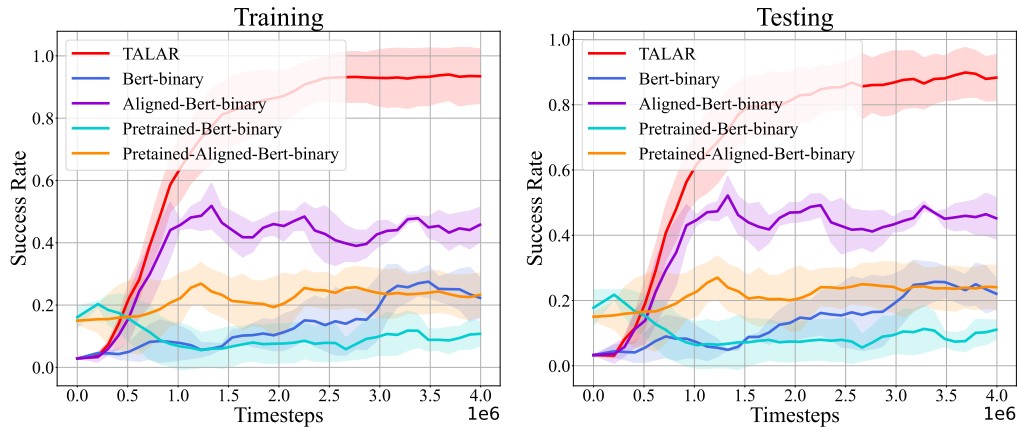

Figure 12: Training curves on FrankaKitchen environments. The new baselines with the same network architecture as TALAR are denoted with the prefix "Aligned".

algorithm. The results are presented in the Fig. 13. The results demonstrate that 10,000 samples is sufficient to train a policy that achieves a success rate of 76%+, which clearly outperforms the performance of other baseline methods. These experimental results suggest that TALAR can effectively train a robust policy even with a limited number of samples in the task dataset.

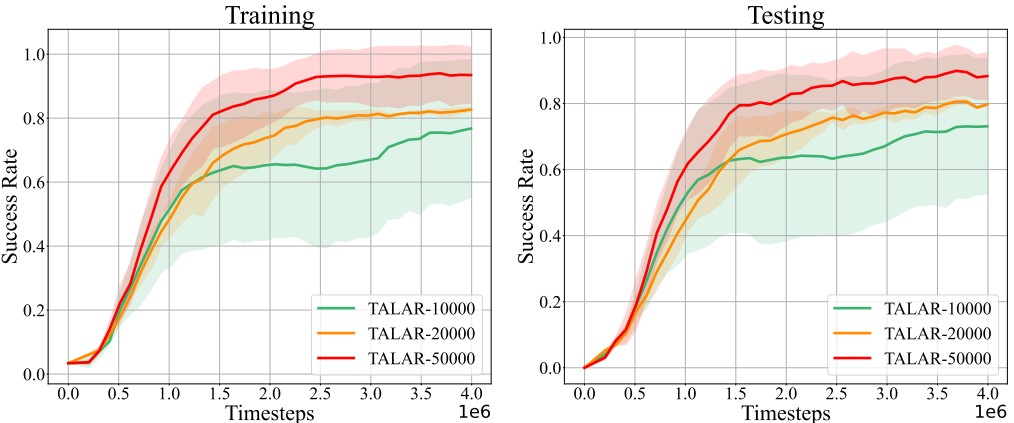

Figure 13: Training curves of TALARtrained on different size of datasets on FrankaKitchen environments. The numbers after TARLAR indicate the size of the dataset used.

### E.5 Fine-tuning BERT-baed baselines

We carried out further experiments where the parameters of the BERT model in baseline methods are updated during the training process, as depicted in Fig. 14. The results indicate that when the parameters of BERT are optimized, the baseline methods struggle to achieve successful task completion. We hypothesize that this outcome may be attributed to the extensive parameter count of the BERT model, which potentially increases the complexity of the learning process.

### E.6 End-to-end TALAR

We conduct additional experiment that trains TL generator and translator jointly on FrankaKitchen task, following the original experiment setting, as shown in the Fig. 15. The experiment results indicate that training the two modules independently is more effective in training a policy.

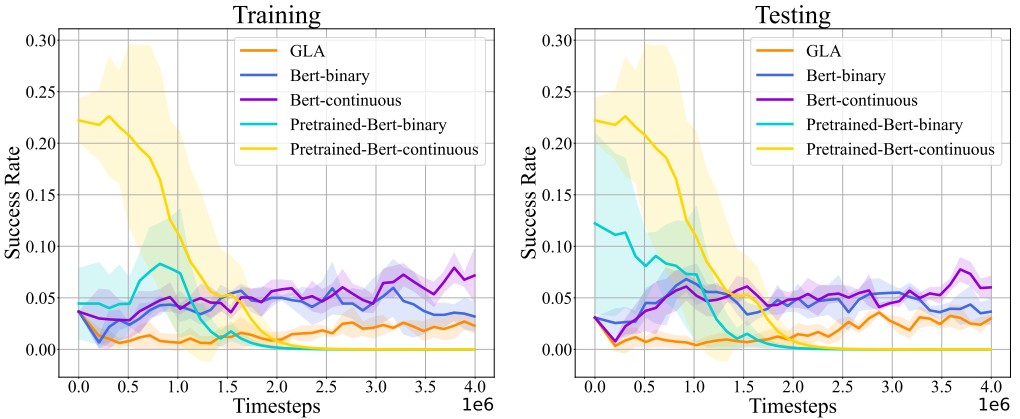

Figure 14: Training curves of fine-tuned baselines on FrankaKitchen environments. The fine-tuned baselines are denoted with the prefix "Pretrained".

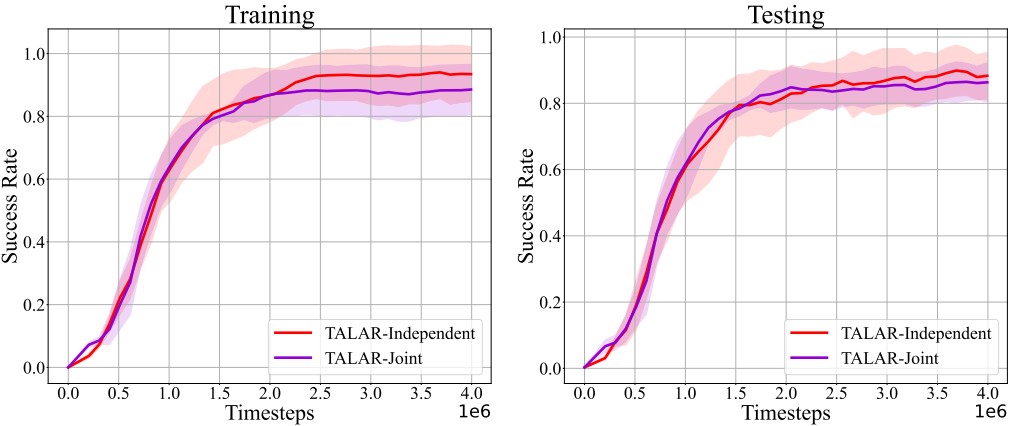

Figure 15: Training curves of end-to-end TALAR on FrankaKitchen environments.

## E.7  Replacing generator with hand-crafted feature

We conduct a supplementary experiment to compare TALAR with a manually crafted structural representation on the FrankaKitchen task. To implement the new baseline, the NL instructions are parsed into a one-hot vector which indicates the current goal-configuration, serving as the structured representation. Subsequently, we implement a translation process same to that of TALAR. All other experimental setting remains consistent with those outlined in our original study. The results of this experiment are presented in the table below, where 'Handcrafted' denotes the new baseline trained based on the handcrafted representation.

The results show that the performance of TALAR is comparable to the handcrafted representation in term of learning speed and the final score (Figure 6). This experiment further justify the effectiveness and conciseness of the task language learned by TALAR.

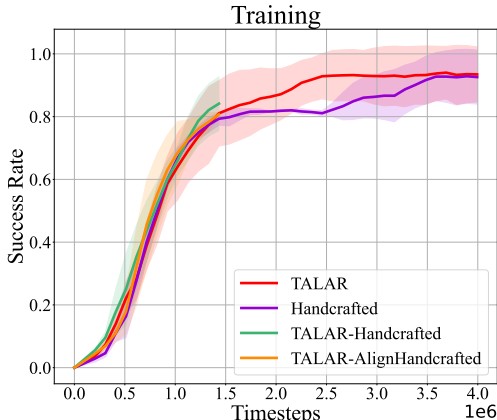

Figure 16: Training curves of hand-crafted features on FrankaKitchen environments.

### E.8 Deployment Examples of IFP Trained by TALAR

We visualize the performance of the IFP trained by TALAR in the FrankaKitchen and CLEVR-Robot environment. Fig. 17 and Fig. 18 presents the instruction-following examples on CLEVR-Robot and FrankaKitchen environments, respectively. The trained IFP demonstrates a strong ability to accurately understand natural language instructions and efficiently accomplish the assigned tasks.

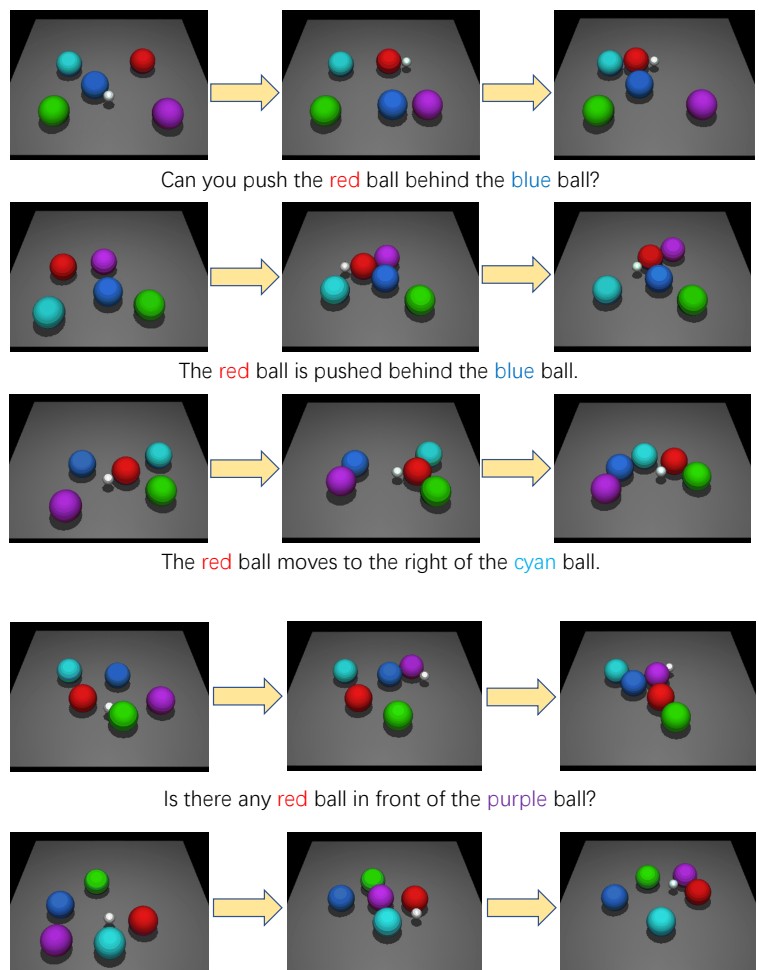

Figure 17: A visualization of the TALAR's IFP deployment process on CLEVR-Robot.

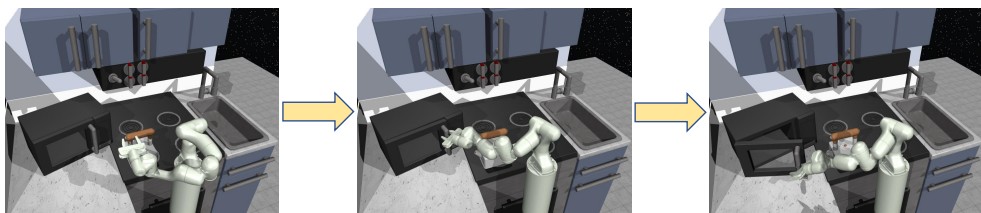

Please open the door of the **microwave**.

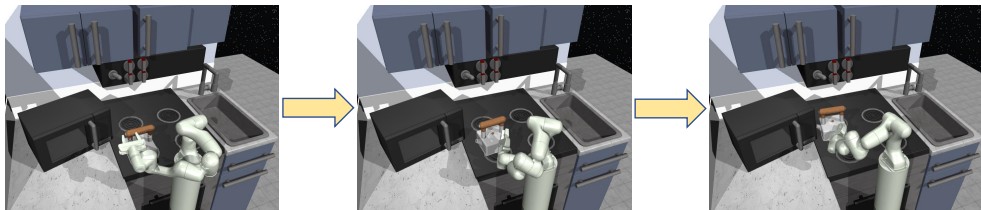

Can you please put the **kettle** on the stove, robot?

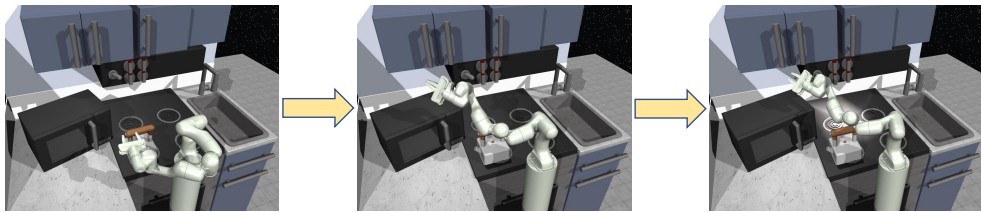

I need some **light**, can you help me?

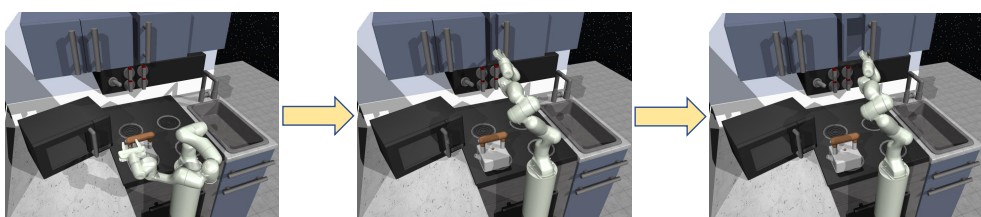

Would you mind opening the **slide cabinet** door?

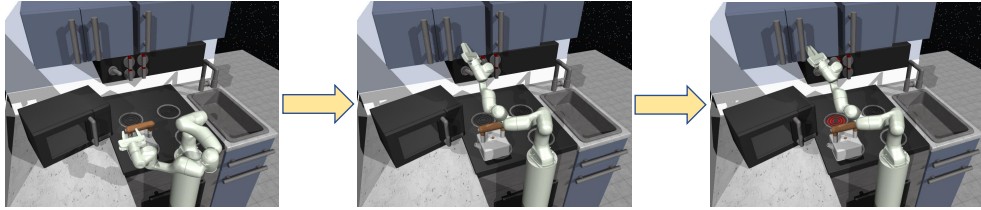

Can you turn on the **top burner** by twisting the oven knob?

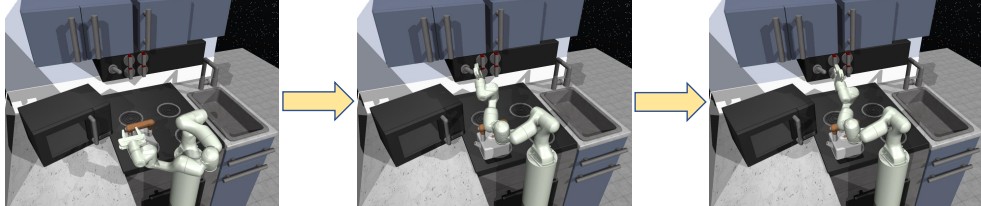

Turn the oven knob to activate the **bottom burner**.

Figure 18: A visualization of the TALAR's IFP deployment process on FrankaKitchen.

# F   Retrievals for Notations and Abbreviations

Table 4: Notations and abbreviations in this paper.

| Name | Meaning |
|------|---------|
| **Notations** | |
| $L_\text{T}$ | task language |
| $L_\text{N}$ | natural language |
| $q_{\phi_1}$ | encoder with parameters $\phi_1$ |
| $p_{\phi_2}$ | decoder with parameters $\phi_2$ |
| $g_\theta$ | TL generator with parameters $\theta$ |
| **Abbreviations** | |
| NL | Natural Language |
| TL | Task Language |
| LM | Language Model |
| RL | Reinforcement Learning |
| IOL | Inside-Out Learning |
| OIL | Outside-In Learning |
| RG | Referential Game |
| GLA | Grounding Language to latent Action |
| NLC-RL | Natural Language-Conditioned Reinforcement Learning |

# G   Societal Impact

This research enhances the success rate and adaptability of instruction-following policies in response to various natural language expressions. The development of NLC-RL agents capable of effectively following human instructions holds the potential to impact multiple aspects of society. One of the most immediate societal impacts lies in improving human-robot collaboration. As these RL agents become increasingly proficient in comprehending and executing human instructions, they can be integrated into various applications, including manufacturing, logistics, and service industries. This integration may lead to heightened efficiency, reduced labour costs, and enhanced safety across these sectors. Moreover, these RL agents' advanced natural language understanding can be applied to assistive technologies for individuals with disabilities. For instance, they could be utilized in intelligent home systems to facilitate navigation for those with mobility impairments or to support individuals with cognitive impairments in accomplishing daily tasks. Nonetheless, it is important to consider the ethical implications of deploying these RL agents in real-world settings. Ensuring that these systems maintain transparency and accountability is of paramount importance.

# Appendix References

[54] Alexander Borgida. On the relative expressiveness of description logics and predicate logics. *Artificial Intelligence*, 82:353–367, 1996.

[55] Angeliki Lazaridou and Marco Baroni. Emergent multi-agent communication in the deep learning era. *arXiv*, 2006.02419, 2020.

[56] M. G. Bellemare, Y. Naddaf, J. Veness, and M. Bowling. The arcade learning environment: An evaluation platform for general agents. *Journal of Artificial Intelligence Research*, 47:253–279, 2013.

[57] Shubham Pateria, Budhitama Subagdja, Ah-Hwee Tan, and Chai Quek. Hierarchical reinforcement learning: A comprehensive survey. *ACM Computing Surveys*, 54(5):109:1–109:35, 2021.

[58] Alexander Sasha Vezhnevets, Simon Osindero, Tom Schaul, Nicolas Heess, Max Jaderberg, David Silver, and Koray Kavukcuoglu. Feudal networks for hierarchical reinforcement learning. In *ICML*, 2017.

[59] Nicholas K Jong, Todd Hester, and Peter Stone. The utility of temporal abstraction in reinforcement learning. In *AAMAS*, 2008.

[60] Ofir Nachum, Haoran Tang, Xingyu Lu, Shixiang Gu, Honglak Lee, and Sergey Levine. Why does hierarchy (sometimes) work so well in reinforcement learning? *arXiv*, 1909.10618, 2019.

[61] Xintong Yang, Ze Ji, Jing Wu, Yu-Kun Lai, Changyun Wei, Guoliang Liu, and Rossitza Setchi. Hierarchical reinforcement learning with universal policies for multistep robotic manipulation. *IEEE Transactions on Neural Networks and Learning Systems*, 33:4727–4741, 2022.

[62] Ofir Nachum, Shixiang Gu, Honglak Lee, and Sergey Levine. Data-efficient hierarchical reinforcement learning. In *NeurIPS*, 2018.

[63] Yiding Jiang, Shixiang Gu, Kevin Murphy, and Chelsea Finn. Language as an abstraction for hierarchical deep reinforcement learning. In *NeurIPS*, 2019.

[64] Karl Pertsch, Youngwoon Lee, and Joseph J. Lim. Accelerating reinforcement learning with learned skill priors. In *CoRL*, 2020.

[65] Antonin Raffin, Ashley Hill, Adam Gleave, Anssi Kanervisto, Maximilian Ernestus, and Noah Dormann. Stable-baselines3: Reliable reinforcement learning implementations. *Journal of Machine Learning Research*, 22:268:1–268:8, 2021.

[66] Rishabh Agarwal, Chen Liang, Dale Schuurmans, and Mohammad Norouzi. Learning to generalize from sparse and underspecified rewards. In *ICML*, 2019.

[67] Geoffrey Cideron, Mathieu Seurin, Florian Strub, and Olivier Pietquin. Higher: Improving instruction following with hindsight generation for experience replay. In *SSCI*, 2020.

[68] Marcin Andrychowicz, Dwight Crow, Alex Ray, Jonas Schneider, Rachel Fong, Peter Welinder, Bob McGrew, Josh Tobin, Pieter Abbeel, and Wojciech Zaremba. Hindsight experience replay. In *NeurIPS*, 2017.

[69] Abhishek Gupta, Vikash Kumar, Corey Lynch, Sergey Levine, and Karol Hausman. Relay policy learning: Solving long-horizon tasks via imitation and reinforcement learning. In *CoRL*, 2019.

