# OpenReview forum: "Natural Language Instruction-following with Task-related Language Development and Translation"
_NeurIPS.cc/2023/Conference — NeurIPS 2023 poster_

### Official Review · Reviewer_rFCy · 2023-06-30

**Soundness:** 3 good
**Presentation:** 3 good
**Contribution:** 2 fair
**Rating:** 5
**Confidence:** 4

**Summary:**

The main contribution of the work is TALAR, a method for learning a vector representation of the instruction via a referential game. In TALAR, a generator creates a task vector from (state, next-state) pairs, which is then back-translated into natural language by the receiver. A translation from natural language to task vectors is then learned. At policy learning time, the instruction gets translated into task language and fed to the policy. The results show that TALAR outperforms other methods of learning task embeddings.


**Strengths:**

Allowing a task representation to emerge in a referential game, and then using the representation in policy learning is original. The experiments are standard in the RL community and show TALAR's superior performance. The finding that a more linearly separable task representation leads to better instruction following performance is significant.


**Weaknesses:**

My main concern with the work is that I don’t understand why the proposed technique would be better than parsing the instruction from humans into a structured representation that’s formulated ahead of time. What is the utility of having the discrete representation of a task emerge, rather than defining a desired discrete representation to begin with? GLA and the BERT methods make more sense to me, because the embedding methods are fairly generic .

It seems to me that you could choose the predicate representation vector ahead of time, instead of playing the referential game, and then do the same translation step as before. Handcrafting another structured representation baseline seems like it would elucidate whether the referential game is necessary. One could do the parsing programmatically, or using an LLM like GPT-3.

Furthermore, what happens when the natural language is OOD in task language? The method doesn’t seem set up to handle distributional shifts, which seems likely because, as the authors say, “natural language is a complex and unbounded representation” (line 29). Because the task representation is both discrete and emergent, is the representation manifold going to be well-behaved for OOD natural language?

I also had some difficulty understanding certain paragraphs in the methods section. See questions.

Comments:
- I recommend including “predicate representation in the title.
- Line 4: I wouldn’t include the terms “outside-in” and “inside-out” in the abstract, because you don’t define them until the intro and the meanings are not immediately intuitive.
- Line 192: Typo “The We”
- Line 194: This paragraph is hard to understand. A receiver diagram would be nice.

My score reflects the fact that after reading the paper, I’m unconvinced on the utility of the core technique in the paper. Happy to discuss this with the authors.


**Questions:**

- Line 197 and Line 203. Are these two sentences contradictory? You say you’re predicting the word corresponding to the state, but the word is chosen randomly?
- Figure 2: is the number of predicate networks linear in the number of objects in the state? How did you choose the number of predicate and argument networks?

**Limitations:**

The discussion is adequate.

---

> ### Author Rebuttal · Authors · 2023-08-10
>
> Thank you for your time and valuable comments. We have taken every question into consideration and revised our paper to fix the typos. Please find the response below.
>
> > Q1: Why the proposed technique would be better than parsing the instruction from humans into a structured representation that’s formulated ahead of time.
>
> A1: Good point. we acknowledge that a manually designed structured representation could potentially be an effective method for interpreting natural language instructions. However, it is important to highlight the advantages of our approach over this traditional method.
>
> - Universality: The primary advantage of our method, TALAR, is its universality. Converting human instructions into a structured representation typically necessitates a unique design for each task, a process that can be both labor-intensive and limited in its ability to manage complex instructions. In contrast, TALAR automatically learns to develop TL and translates NL to TL, thereby eliminating the need for manual parsing of natural language instruction representation. This learning process is both automatic and universal.
> - Scalability: TALAR's task language is designed to adapt and scale in accordance with the complexity of the environment and the tasks that agents are required to perform. This feature proves particularly beneficial in scenarios involving complex tasks with numerous objects and operations, where creating a structured representation for natural language instructions can be challenging. In contrast, our method learns task language in an unsupervised manner through engagement in a referential game with the receiver. This learning framework can be easily applied to various NLC-RL tasks.
>
> > Q2: Handcrafting another structured representation baseline seems like it would elucidate whether the referential game is necessary.
>
> A2: We conduct a supplementary experiment to compare TALAR with a manually crafted structural representation on the FrankaKitchen task. To implement the new baseline, the NL instructions are parsed into a one-hot vector which indicates the current goal-configuration, serving as the structured representation. Subsequently, we implement a translation process same to that of TALAR. All other experimental setting remains consistent with those outlined in our original study. The results of this experiment are presented in the table below, where ‘Handcrafted’ denotes the new baseline trained based on the handcrafted representation.
>
> |  | TALAR | Handcrafted |
> | --- | --- | --- |
> | Success rate | 93.5±8.8 | 94.1±8.1 |
>
> The results show that the performance of TALAR is comparable to the handcrafted representation in term of learning speed and the final score (Figure 6). This experiment further justify the effectiveness and conciseness of the task language learned by TALAR.
>
> > Q2: Furthermore, what happens when the natural language is OOD in task language?
>
> A2: At present TALAR handles with OOD natural language based on the generalization ability of translator. The translator's language model is trained on extensive corpus data, thereby demonstrating a certain degree of generalization ability. Preliminary experiments on unseen natural language instructions, although potentially not sufficiently OOD, provide initial evidence of this generalization ability.
>
> Simultaneously, we recognize that managing OOD datasets is a valuable area of exploration. A direct yet potentially effective strategy could involve designing an OOD detection module and processing the OOD natural language instructions differently. For instance, we could consider updating the translation generator or translator module online.
>
> > Q3: I also had some difficulty understanding certain paragraphs in the methods section.
>
> A3: Apologize for the method description not being clear enough. Here we would like to take this opportunity to clarify the particular details of our method.
>
> **Comment1**: Figure 2: is the number of predicate networks linear in the number of objects in the state? How did you choose the number of predicate and argument networks?
>
> Response1: The number of predicate networks is not necessarily linear to the number of objects in the state. It serves as a representation of the potential relationships that our learning algorithm might identify within a specific task. These relationships can often be abstract and anonymized, which makes their exact quantification a complex endeavor. Thus, the selection of the number of predicate networks is mainly influenced by the scale of the task, which includes factors such as the complexity of the instructions, the number of objects. In our experiment, we opted for a moderate number of 4 for both the FrankaKitchen and CLEVR-Robot tasks. Our ablation study on the number of predicate networks (Figure 7 in the original paper) show that TALAR performs consistently well across different selections of predicate network numbers.
>
> **Comment2**: Are two sentences at line 197 and line 203 contradictory?
>
> Response2: Sorry for the unclear presentation. The sentence on line 197 provides a high-level overview of the receiver's training objective, while the sentence on line 203 delves into the specifics of our implementation, and they are not contradictory.
>
> In the context of the $(s,s’,L_N)$ tuple, we initially select a word $T_i$ at random from the $L_N$ sentence. Subsequently, the receiver's goal is to predict this selected word, taking into account the task language (produced by the TL generator) and the previous words (i.e., $T_{1,\cdots,i-1}$). This process ensures that the TL generator is important to generate effective task language, as its effectiveness directly influence the receiver's prediction accuracy.
>
> **Comment3**: This paragraph is hard to understand. A receiver diagram would be nice.
>
> Response3: We have incorporated a receiver diagram to enhance the clarity of the discussed concept, which is depicted in Figure 7 in the rebuttal attached PDF file.

---

> > ### Comment · Reviewer_rFCy · 2023-08-14
> >
> > Thank you for A1 and A2. These answers get at my main concerns of TALAR's usability, and I have increased my score. I have reservations as to whether this solution is better than a semantic parsing solution, especially since semantic parsing is so good nowadays with LLMs, so I do not increase my score more.
> >
> > Nevertheless, after reading the rebuttal I think the work is technically correct.

---

> > > ### Author Response · Authors · 2023-08-14
> > > **Response to follow-up comments**
> > >
> > > Thank you for your follow-up comments. We appreciate your feedback and are pleased to note that our response has addressed your main concerns.
> > >
> > > ------
> > > Regarding your follow-up comment:
> > >
> > > > Whether this solution is better than a semantic parsing solution, especially since semantic parsing is so good nowadays with LLMs.
> > >
> > > We acknowledge your concern and would like to discuss the related studies on semantic parsing using LLMs. While it is true that LLMs can generate the semantic representation of a task, prior research [1] has underscored several challenges associated with LLM-based methods. These include (1) the complexity of semantic decomposition, (2) the necessity for specific prompt design to generate semantic representation, and (3) the potential for the knowledge required for translation to exceed the capacity of a single prompt. As a contrast, our method provides an effective way to automatically generate task representation.
> > >
> > > Moreover, our method holds a distinct advantage in terms of model size: Efficient semantic parsing using LLMs may require language models with an extensive number of parameters, and there is a significant performance gap (as demonstrated in paper [3]) between models with 300+M parameters and those with 10+B parameters. To employ LLM for task-specific semantic parsing, [1] utilizes code-davinci-002 and [2] employs codex as the LLM for parsing. All these LLMs possess billions of parameters. In contrast, our method, which employs a BERT model (110M parameters) to encode natural language sentences, offers a more efficient parsing solution.
> > >
> > > ------
> > >
> > > Thanks again for your time and effort in providing feedback. We are happy to discuss if you had any additional concerns.
> > >
> > >
> > > ### Reference:
> > > [1] Andrew Drozdov, et al. Compositional Semantic Parsing With Large Language Models. ICLR 2023.
> > >
> > > [2] Zhoujun Cheng, et al. Binding Language Models in Symbolic Languages. ICLR 2023.
> > >
> > > [3] Erik Nijkamp, et al. Codegen: An Open Large Language Model for Code With Multi-Turn Program Synthesis. ICLR 2023.

---

### Official Review · Reviewer_KoSw · 2023-07-05

**Soundness:** 4 excellent
**Presentation:** 4 excellent
**Contribution:** 3 good
**Rating:** 6
**Confidence:** 3

**Summary:**

This paper proposes the framework of TALAR, an Inside-Out learning framework for training policies that follows language instructions with reinforcement learning. The method leverages predicate representation for building a compact space of task language, and learns the generator of the task language from the game state by reconstructing natural language. A translator from natural language to task language is learned by language translation loss, and a policy based on task language is learned by reinforcement learning. In experiment, TALAR shows better performance than baseline methods that use natural language embeddings on two continuous control environments.

**Strengths:**

This paper has overall great presentations of the motivation, methods, and experiment.

TALAR has the clear motivation of learning the task-correlated language and optimizing a policy based on task language, instead of using generally pre-trained language embeddings by language models.

The experiments in section 5.2 show that TALAR learns better embedding space compared to BERT pre-trained models.

**Weaknesses:**

Some baseline methods are not compared: method using the same network architecture or number of parameters as TALAR but with no Task Language translation loss, trained with behavior cloning and reinforcement learning. This will alleviate the concerns that the performance improvement is caused by additional network capacity introduced by the autoregressive translation, while the baseline methods only have a very small number of parameters being tuned (the FC layer after BERT, if it is the case).

**Questions:**

In figure 2(b), it seems that there is no mechanism for selecting different arguments for different predicates. Do all the predicates have the same set of arguments? If so, since the predicate is a Boolean variable, it is reasonable to suspect that that the multiple predicates cannot convey rich information in the arguments.

In line 222, it is not precisely described how multiple predicates are concatenated to form the task language.

**Limitations:**

The limitations are well addressed in the conclusion section.

---

> ### Author Rebuttal · Authors · 2023-08-10
>
> We appreciate your time and effort in reviewing our paper and providing valuable feedback. We are glad that you found our presentation, soundness, and the TALAR method to be strong. We would like to address your concerns and questions as follows.
>
> > Q1: Some baseline methods are not compared: method using the same network architecture or number of parameters as TALAR but with no Task Language translation loss.
> >
>
> A1: We have incorporated a comparison with these baselines, which share the same network architecture as TALAR. More specifically, we have added new BERT-based baselines on FrankaKitchen task, which employ the same architecture as the TL generator in TALAR, adhering to the BERT model. Other experimental settings remain same with the experiments in the paper. These new baselines are denoted with the prefix 'Aligned', and their respective experimental results are presented in the following table:
>
> |  | TALAR | Bert-binary | Aligned-Bert-binary | Pretrained-Bert-binary | Aligned-Pretrained-Bert-binary |
> | --- | --- | --- | --- | --- | --- |
> | Training | **93.5±8.8** | 22.3±6.6 | 45.8±5.6 | 10.8±4.1 | 23.3±7.0 |
> | Testing | **88.3±7.1** | 22.1±4.8 | 45.2±6.3 | 11.1±3.0 | 24.1±6.8 |
>
> The performance of the baseline improves with the implementation of a new network architecture. However, it remains inferior to the TALAR approach in terms of learning efficiency and convergence scores. This highlights the effectiveness of the TALAR method. Please refer to Figure 4 in the rebuttal attached PDF file for the training curves. We appreciate your suggestions about the network architecture of baselines, and would add the new experiment results to the revised version.
>
> > Q2: In figure 2(b), it seems that there is no mechanism for selecting different arguments for different predicates. Do all the predicates have the same set of arguments?
> >
>
> A2: At present, we have implemented the TL generator such that all predicate networks utilize a shared argument list. This approach proves to be both sufficient and effective when the task language can be represented using a limited number of arguments. However, the TALAR system is inherently flexible and can be readily extended to accommodate more complex tasks. For instance, we can enable each predicate network to input a unique argument list by constructing additional, separate argument networks.
>
> > Q3: In line 222, it is not precisely described how multiple predicates are concatenated to form the task language.
>
>
> A3: We apologize for unclear presentation in the original paper regarding these points. The resulting task language is represented as $[Pred_1,\cdots,Pred_{N_{pn}}, arg_1, \cdots, arg_{N_a}]$, where $\text{Pred}_{i}$ denotes the output of the i-th predicate network. To enhance clarity, we will incorporate the explanation to Section 4.2.1.
>
> We hope that our response has addressed your concern and questions satisfactorily. If you had any further concerns, we are glad for discussion.

---

> > ### Comment · Reviewer_KoSw · 2023-08-21
> > **Response to the rebuttal**
> >
> > Thanks to the authors for the detailed response. I appreciate the additional results and clarifications from the authors.

---

> > > ### Author Response · Authors · 2023-08-21
> > > **Thanks for your follow-up comments**
> > >
> > > Thanks for your positive feedback and follow-up comments. We are pleased to note that the additional results and clarifications were useful in addressing your concerns.

---

### Official Review · Reviewer_nXhV · 2023-07-10

**Soundness:** 3 good
**Presentation:** 2 fair
**Contribution:** 3 good
**Rating:** 6
**Confidence:** 3

**Summary:**

This paper presents an algorithm that reduces the policy learning burden in a natural language-conditioned reinforcement learning framework. The authors investigate an inside-out scheme for natural language-conditioned RL and then present a new approach, TALAR, that learns multiple predicates to model object relationships as the task language. In the experiments, the authors demonstrated that TALAR outperforms previous baseline algorithms with instruction following policy.

**Strengths:**

- This proposed method helps policy learning by learning representations so that natural language instructions are close to semantically similar ones.
- It shows much better performance than baseline algorithms in various domains.

**Weaknesses:**

- A separate training dataset is required for TALAR learning, and the cost for this seems expensive.
- NL instructions must be defined separately for each task and data must be collected.
- Performance is sensitive depending on the type of NL instructions collected for training.
- It is difficult to create a training dataset including NL instructions without prior knowledge of the domain.

**Questions:**

- How to construct $s$ and $s’$ of $(s,s’,L_N)$ in task dataset $D$? As mentioned in lines 283-285, when composing $(s, s’, L_N)$, all $s$ from timestep 0 to $T-1$ are used, but doesn’t most of them correspond only to the partial part of $L_N$, not exactly $L_N$?
- It seems that annotation for predicate representation is also necessary for referential game learning, is that correct? If so, isn't the amount of information required different from other baseline algorithms? It would be better if the annotations required for each algorithm were specified.
- In lines 256-258, it is mentioned that the NL instruction was created using ChatGPT. How did you create it using it specifically? It would be better to provide a more detailed explanation with an example.

**Limitations:**

The authors mentioned limitations in their paper.

---

> ### Author Rebuttal · Authors · 2023-08-09
>
> We appreciate your insightful comments and constructive feedback on our paper. We are grateful for the time and effort you put into reviewing our work. Below we address each of your concerns and questions.
>
> > Q1: A separate training dataset is required for TALAR learning, and the cost for this seems expensive.
> >
>
> A1: We appreciate your concern regarding the potential expense associated with the need for a separate training dataset for TALAR learning.
>
> We would like to clarify that the dataset is necessary for connecting the RL agent with the natural language. While some baseline methods do not explicitly use a dataset containing natural language, the language model they depend on also require amounts of corpus data to be trained.
>
> Besides, the acquisition of this data does not inherently imply a high cost, as the dataset for TALAR learning does not necessarily have to be generated from scratch. Existing datasets can be effectively repurposed or augmented to meet our requirements. For instance, pre-collected datasets have been extensively utilized in the realms of offline RL, imitation learning, and language model training.
>
> We recognize that the size of task dataset is a significant factor to consider. Thus we have conducted additional experiments to investigate the impact of task sample number on the performance of the algorithm. The results are presented in the Figure 3 in the rebuttal PDF file. The results demonstrate that 10,000 samples is sufficient to train a policy that achieves a success rate of 76%+, which clearly outperforms the performance of other baseline methods. These experimental results suggest that TALAR can effectively train a robust policy even with a limited number of samples in the task dataset.
>
> > Q2: NL instructions must be defined separately for each task; It is difficult to create a training dataset including NL instructions without prior knowledge of the domain; Performance is sensitive depending on the type of NL instructions collected for training.
> >
>
> A2: We would like to clarify that these instructions are not 'defined' in any rigid sense. In practical, the NL instructions in the task dataset could be flexibly assigned based on the linguistic habits of the person providing the state transition descriptions. This means that we do not require specific design of NL instructions to align with real-world application scenarios.
>
> To illustrate, while constructing the task dataset, one could generate NL instructions by observing a video of a robotic manipulation (i.e., the trajectory) and then describing the robot's execution goal in their own words. This process does not necessitate prior knowledge of the task domain. For example, we use ChatGPT to generate the NL instructions to construct the task dataset. Our experiments in Figure 4 in the original paper have shown that TALAR is robust to unseen NL instructions (please refer to Appendix D.1.1 for examples of training/testing NL instructions). We believe the experimental results underscore the robustness of the proposed approach.
>
> > Q3: How to construct $s$ and $s’$ of $(s,s’,L_N)$ in task dataset $D$?
> >
>
> A3: In practical, we could pre-collect the trajectory data $\{s_0,s_1,\cdots,s_T\}_i$, and have a human observer describe the instruction for each trajectory in natural language. Ideally, we would use $(s_0, s_T)$ to construct the task dataset. However, this could potentially make the data collection process both costly and labor-intensive. To mitigate this issue, we utilize a data augmentation strategy. We take the intermediate state in the trajectory as $s$, and the terminal state as $s’$. This approach treats NL instructions similarly to the goal in a goal-conditioned RL setting, where all states in the trajectory share the same goal. Consequently, it is sufficient for the agent to determine its action based on the natural language instruction and current state. We believe this data augmentation strategy not only enhances the efficiency of data collection but also enriches the diversity of the task dataset, thereby improving the robustness and generalizability of our model.
>
> > Q4: It seems that annotation for predicate representation is also necessary for referential game learning.
> >
>
> A4: Good point. While we acknowledge that the annotation for predicate representation might potentially accelerate the learning of task language, these annotations are not necessary.   There are two primary considerations.
>
> Firstly, the central objective of our research is not to match the manually designed NL representations for each NLC-RL task. Instead, our focus is on the automatic discovery of task-specific relationships, which we refer to as task language. This approach aligns with our broader aim of fostering a more autonomous and adaptable system.
>
> Secondly, providing predicate representation could be laborious and costly. In such scenarios, the benefits of our approach become particularly evident. Our system learns predicate networks in an unsupervised manner through engagement in a referential game with the receiver.
>
> > Q5: How did you use ChatGPT specifically?
> >
>
> A5: The specific prompt we use is as follows: "Suppose you want to order a home service robot to [task description]. Give me 100 kinds of diverse expressions. Note that you don't need emphasize you are talking with a robot and don't use word ‘robot’." Here, [task description] is a placeholder for the specific task, such as 'open the microwave door'.
>
> We hope that these responses can address your concerns and questions. If you had any further concerns, please let us know.

---

> > ### Comment · Reviewer_nXhV · 2023-08-20
> > **Response to the rebuttal**
> >
> > Thank you for taking your time to respond to my review. I read the authors' rebuttal and other reviews. Since the authors have addressed most of my questions and concerns, I would like to raise my score. I thank the authors for their response.

---

> > > ### Author Response · Authors · 2023-08-20
> > >
> > > We are encouraged that our response has addressed most of your questions and concerns. We appreciate the time and effort you have invested in providing insightful feedback.

---

### Official Review · Reviewer_rdPR · 2023-07-24

**Soundness:** 4 excellent
**Presentation:** 4 excellent
**Contribution:** 3 good
**Rating:** 7
**Confidence:** 4

**Summary:**

This work tackles the problem of learning an effective natural language (NL) instruction-following agent using RL algorithm as a goal-conditioned RL setup. The authors propose to learn a task-language (TL), which is a synthetic and vectorized representation containing the abstractive meaning of the instruction, via a referential-game-like manner. A TL generator alongsidse an NL-to-TL translator is learned for deriving an expressive and concise representation for an RL algorithm to effectively utilize. The experiments conducted on simulated environment demonstrate superior performance compared to several instruction-following baselines, as well as better distributed latent representations for the goal-conditioned RL policy.

**Strengths:**

- It is an interesting idea to position referential game in developing the TL, and the efficacy of such a language is the key to effective communication during the guided RL.
- The proposed framework is well-modularized and component-wise it can be interpretable to certain extent for model analysis.
- The paper presentation is easy to follow, with nicely illustrated visualizations.
- Experiments are solid, and the supplementary materials are helpful.

**Weaknesses:**

- Could you elaborate the rationale behind learning the TL independent to a translator? (I’m guessing better modularization or something similar.) The TL developed this way is not expected to generalize very well across many tasks, and sacrifices the expressiveness of NL to a specifically trained translator. I.e., why not learn the translator (and the TL) in an end-to-end fashion (such as VQ-VAE paradigm or methods like discrete code-book look up), and maybe compare its pros and cons to the proposed framework? (The training can utilize bascially the same settings as the proposed framework in this work.)
- Following above, if my understanding is correct, the objective of the “receiver” is something like next-word prediction or MLM in BERT. In this sense, the L_T operates similar to a prefix. How would you ensure it is indeed conditioned on the prefix for the word prediction? It is intuitive to think that the model will anyway predict the word correctly if trained well under standard language modeling objective and/or MLM.
- A discussion in comparisons with recent robotic works that utilize LLM as a strong resource/planner is needed, such as [1].
- A set of non-frozen LM baselines could be conducted to even further emphasize on the needs of the conciseness of the generated TL.

[1] Ahn, Michael, et al. "Do as i can, not as i say: Grounding language in robotic affordances." arXiv 2022.

**Questions:**

- How does it compare to program synthesis lines of research? PL can be an expressive and logical (procedural), which contain deterministic execution routes. And PL can be more expressive than the utilized predicate representation for more complex and long horizon tasks. I think a discussion could be nice. [2] [3]
- Referring to the corresponding weakness, how would authors envision/suggest future works to apply similar (and/or extended) paradigm to more complex instruction following tasks, such as navigation, robot manipulation, etc., that are guided with much more sophisticated language?
- Is L_T^tilt in Section 4.2.2 discrete or continous?
- Better to include the prompting scheme for ChatGPT instruction generation (basically, paraphrases).

[2] Sun, Shao-Hua, et al. "Neural program synthesis from diverse demonstration videos." ICML-18.
[3] Sun, Shao-Hua, Te-Lin Wu, and Joseph J. Lim. "Program guided agent." ICLR-19.

**Limitations:**

- Please refer to the weaknesses for the limitations of this work.

---

> ### Author Rebuttal · Authors · 2023-08-09
>
> Thank you for carefully reviewing our paper and providing constructive comments. We hope that our response has addressed your concerns, but if we missed anything please let us know.
>
> > Q1: Could you elaborate the rationale behind learning the TL independent to a translator?
> >
>
> A1: The main reason we separate TL development and translation is we can handle and expand these two modules independently. For example, developing TL during the RL training phase.
>
> Besides, we consider the practical aspects of the training procedure. Training these two modules concurrently could potentially lead to a trivial solution (e.g., both the generator and translator outputting zero to achieve a low translation loss). Furthermore, the back-propagation of loss from the translator could potentially interfere with the effective development of the TL.
>
> To better illustrate, we conduct additional experiment that trains TL generator and translator jointly on FrankaKitchen task, following the original experiment setting, as shown in the Figure 1 in the rebuttal PDF file. The experiment results indicate that training the two modules independently is more effective in training a policy.
>
> > Q2: The objective of the “receiver” is something like next-word prediction or MLM in BERT. How would you ensure it is indeed conditioned on the prefix for the word prediction?
>
> A2: Yes, the training objective of receiver has similar form like MLM loss in BERT, while it aims at predicting the next token based on the task language and previous tokens. In some cases, the tokens cannot be accurately predicted without the prefix. For example, there could be similar natural language instructions like “Open the refrigerator door” and “Open the microwave door”. The generated $L_{\rm T}$ must capture the key information in the state pair to assist model in correctly predicting ‘refrigerator’ token.
>
> > Q3: A discussion in comparisons with recent robotic works that utilize LLM as a strong resource/planner is needed.
>
> A3: We would like to discuss related works about language-based robotics control [1,2,3]. SayCan [1] combines low-level tasks with LLMs so that the language model provides high-level knowledge about the procedures for performing complex and temporally extended instructions. Inner Monologue [2] makes further improvements by adding the eponymous “inner monologue", which is implemented as injected feedback from the environment. ReAct [3] introduces LLMs reasoning to help the model induce, track, and update action plans as well as handle exceptions. Overall, these works design methods that generate high-level language instructions, while our work provides efficient ways to learning to complete the instructions.
>
> > Q4: A set of non-frozen LM baselines could be conducted.
>
> A4: We conducted additional experiments where the parameters of the BERT model in baseline methods are updated during the training process, as depicted in Figure 2 in the rebuttal PDF file. The results indicate that when the parameters of BERT are optimized, the baseline methods struggle to achieve successful task completion. We hypothesize that this outcome may be attributed to the extensive parameter count of the BERT model, which potentially increases the complexity of the learning process.
>
> > Q5: How does it compare to program synthesis lines of research?
>
> A5: That’s a very interesting question and we think that there are two key differences:
>
> 1. the expressiveness and complexity of the target domain-specific language (DSL). Program synthesis often deals with general-purpose programming languages, such as Python, which have rich syntax and semantics. TALAR uses a simpler and more restricted DSL. The advantage of using a simpler DSL is that it can be easier to learn and understand by the agent and achieves high learning efficiency. A potential limitation is that it may not fully encapsulate all the subtleties and variations inherent in natural language instructions.
> 2. the way the natural language instructions are translated into the DSL. Program synthesis typically depends on semantic parsing techniques, which necessitates extensive linguistic knowledge and domain expertise, and it can be challenging to manage ambiguity, noise, or incompleteness in natural language. In contrast, TALAR employs a neural network-based translator to map natural language to the task language, thereby eliminating the need for prior knowledge about the task.
>
> > Q6: How would authors envision/suggest future works to apply similar (and/or extended) paradigm to more complex instruction following tasks?
>
> A6: One possible direction is to develop a more expressive and flexible task language. Currently TALAR only employs the format of predicate representation as the task language. How to improve the inherent property of the learned task language is a key problem to be solved. For example, we could add some constraints to the learning process to ensure the TL generator could learn symmetric predicate, which makes it more meaningful.
>
> > Q7: Is $\tilde{L_{T}}$ in Section 4.2.2 discrete or continuous?
>
> A7: $\tilde{L_{T}}$ outputted by translator is **discrete**.
>
> > Q8: the prompting scheme for ChatGPT instruction generation.
>
> A8: The specific prompt we use is as follows: “Suppose you want to order a home service robot to [task description]. Give me 100 kinds of diverse expressions. Note that you don't need emphasize you are talking with a robot and don't use word ‘robot'.” Here, [task description] is a placeholder for the specific task, such as 'open the microwave door'.
>
> ### References:
>
> [1] Ahn, Michael, et al. Do as i can, not as i say: Grounding language in robotic affordances.
>
> [2] Wenlong Huang, et al. Inner monologue: Embodied reasoning through planning with language models.
>
> [3] Shunyu Yao, et al. ReAct: Synergizing Reasoning and Acting in Language Models.
>
> [4] Shao-Hua Sun, et al. Neural program synthesis from diverse demonstration videos.
>
> [5] Shao-Hua Sun, et al. Program guided agent.

---

> > ### Comment · Reviewer_rdPR · 2023-08-10
> >
> > Thanks for the responses, majority of them are answered.
> >
> > For Q2, the reason might because the instructions are relatively short, so those two tools have almost equal probability to be predicted. I agree with this case. However, it could be just more beneficial to learn to model those key entities instead of standard MLM and hoping the models may hit them frequently by  chance.
> >
> > Q3, I'm not sure if I'd agree, SayCan also needs to execute the generated plan. What I would argue for this work is perhaps that the TL is more straightforward (empirically though) to the goal-conditioned learning module. But this could be true as well to those goal-driven RL using just some simple language. In the extreme case, I'd even argue that generating the exact ROS programs are just better but an intermediate symbolic representation like this one could be a nice alternative.
> >
> > Q4, that is an interesting observation. Although it may be beyond the scope of this work, more research should be done on incorporating model update for goal-driven RL using LMs.
> >
> > Q5, I think as long as your target language has a deterministic representation and a compiler that execute it (deterministically), they are programs (doesn't have to be real-world programming language). So, in a sense, TALAR is also doing something similar to program synthesis, but not in the actual DSL token domain. I do like the point on eliminating the domain expertise for target PL but I don't think that's entirely the case here, as even knowing there has to be a predicate and entity -- is a domain knowledge. While it doesn't diminish the contributions of this work, what I expected was more like discussing how this could potentially benefit (or benefit from) the synthesis community and ease some domain overhead.

---

> > > ### Author Response · Authors · 2023-08-11
> > > **Response to follow-up questions**
> > >
> > > Thanks very much for your valuable response. We response to the follow-up questions as follow.
> > >
> > > ------
> > >
> > >
> > > > 1. It could be just more beneficial to learn to model those key entities instead of standard MLM.
> > >
> > > Thanks for your suggestions and we agree with this point. We propose TALAR as one specific implementation of our proposed IOL framework, and we are willing to investigate the potentially more effective implementation. A preliminary idea is, we might develop TL during the RL phase, and the learning objective of referential game becomes the successful task completion, learning latent representation of trajectory, or something else.
> > >
> > > > 2. Discussions about SayCan and robotics works utilizing LLMs.
> > >
> > > We appreciate your comments on the lines for robotics works using LLMs as planner. We will incorporate the discussions on these works into Section 2 (Related Work).
> > >
> > >
> > > > 3. more research should be done on incorporating model update for goal-driven RL using LMs.
> > >
> > > The inefficiency of non-frozen parameter baselines in learning may be attributed to several factors. These could include the quantity of network parameters, the variety of LMs, the extent of NL instructions, and the complexity of RL tasks. We would like to investigate these potential causes to facilitate the effective and efficient construction of natural language-conditioned agents.
> > >
> > > > 4. While it doesn't diminish the contributions of this work, what I expected was more like discussing how this could potentially benefit (or benefit from) the synthesis community and ease some domain overhead.
> > >
> > >
> > >
> > > We appreciate your feedback and clarification on the program synthesis perspective.
> > >
> > > One possible benefit of TALAR for the synthesis community is that it could provide a new way of generating programs from natural language that does not rely on semantic parsing or rule-based methods. For example, utilizing language model and neural networks to learn to generate PL from data (maybe we could prepare some task-specific descriptions/constraints in the dataset for training the neural network), which could enable more robust and efficient program synthesis for various domains and tasks.
> > > On the other hand, the program synthesis techniques could inspire new designs of task language. For example, program analysis techniques could be employed to automatically infer or optimize the structure and semantics of the task language.
> > >
> > > ------
> > >
> > > Thanks again for the careful and timely response. We are glad to any further discussions.

---

### Author Rebuttal · Authors · 2023-08-10

We would like to express our gratitude to the reviewers and chairs for their valuable time and constructive feedback on our paper. We have carefully considered each comment and provided detailed responses. Thanks to the insightful assessments from the reviewers, we have conducted a more thorough exploration about our method (Reviewer rdPR, nXhV), performed a comprehensive comparison with baselines (Reviewer KoSw, rFCy), further highlighted the effectiveness of our method (Reviewer rdPR, KoSw, rFCy), and improved the presentation regarding predicate representation (Reviewer rdPR, nXhV, KoSw, rFCy). The results of these experiments can be found in the attached PDF file.

We also would like to clarify the advantage of our proposed IOL framework and its implementation, TALAR. Our framework has the capability to automatically develop task-specific language, which can subsequently be utilized in the RL phase to enhance the learning efficiency of the RL agent. In comparison to the baseline method, which utilizes a pretrained language model to encode natural language instructions, our approach produces more concise task language, enabling more efficient policy learning. Additionally, unlike methods that require a pre-defined structured representation, our approach eliminates the need for task-specific design, reducing the need for laborious efforts.

We hope our response could address your concerns about our paper. Please let us know if you have any further questions or concerns.

---

### Decision · Program_Chairs · 2023-09-21

**Decision:**

Accept (poster)

**Comment:**

This work proposes conditioning natural language (NL) instruction-following policies on a “task language” (TL) rather than on the raw NL input. The raw NL is translated to TL, which allows for efficient conditioning that leads to more robust performance. The reviewers generally think that this is a good approach and the author rebuttal alleviated several concerns.

I agree with reviewer rFCy that I would like to see a more explicit comparison to and discussion of semantic parsing (which as the author rebuttal remarks can be a useful comparison since their TL approach might be more efficient as it can specialize to the task much more). The idea of having an intermediate structured representation is not new and I think the paper needs to do a much better job at acknowledging this.